# Development of an ultrahigh affinity, trimeric ACE2 biologic as a universal SARS-CoV-2 antagonist
Juliet Gonzales [1,2,4], Tynan Young [2,4], Hyeran Choi[2], Miso Park[2], Yead Jewel[2], Chengcheng Fan[3], Rahul Purohit[2], Pamela J. Bjorkman [3] & John C. Williams [1,2] ✉

Severe acute respiratory syndrome coronavirus 2 (SARS-CoV-2), responsible for the COVID-19 pandemic, utilizes membrane-bound, angiotensin-converting enzyme II (ACE2) for internalization and infection. We describe the development of a biologic that takes advantage of the proximity of the N-terminus of bound ACE2 to the three-fold symmetry axis of the spike protein to create an ultrapotent, trivalent ACE2 entry antagonist. Distinct disulfide bonds were added to enhance serum stability and a single point mutation was introduced to eliminate enzymatic activity. Through surface plasmon resonance, pseudovirus neutralization assays, and single-particle cryo-electron microscopy, we show this antagonist binds to and inhibits SARS-CoV-2 variants. We further show the antagonist binds to and inhibits a 2003 SARS-CoV-1 strain. Collectively, structural insight has allowed us to design a universal trivalent antagonist against all variants of SARS-CoV-2 tested, suggesting it will be active against the emergence of future mutants.

The COVID-19 pandemic, driven by SARS-CoV-2 in late 2019, has had a profound impact on human health. As of April 2024, over 770 million people had been infected and over 7 million succumbed to the disease worldwide[1,2]. Remarkably, vaccines were produced in record time and are now widely available, providing protection and dramatically reducing death rates[3–5]. Despite these efforts, the virus continues to spread for numerous reasons, including variable immune responses and vaccine hesitancy[6]. Critically, uneven measures to eliminate infection have acted as selective pressure[7–9] and provided the virus ample opportunity to evolve[7]. Although SARS-CoV-2 has a lower mutation rate[10] compared to other RNA viruses (e.g., HIV[11]), each new variant has the potential to escape current therapeutics and/or reduce the efficacy of vaccines. The Omicron variant, for instance, encodes over 50 known amino acid mutations[12] in the spike protein compared to the original Wuhan-Hu-1 strain[13–15].

Mortality from COVID-19 dramatically increases with age[16,17] and for those with compromised immune systems[18–20]. A study comparing COVID-19-related hospitalizations, conducted from January 2022 to December 2022, found that immunocompromised individuals comprised 3.9% of the study population, but accounted for 24% of COVID-19-related deaths[21].

As SARS-CoV-2 variants arise that are more antigenically distinct from Wuhan-Hu-1 strain, current vaccines are becoming less potent[22,23]. A study that assessed 50 mAbs against the initial strain and tested against Omicron sub-variants BA.1/BA.1.1/BA.2/BA.3 found that 37 mAbs failed to neutralize the virus, and the 13 remaining mAbs had decreased potency[24]. Furthermore, as both SARS-CoV-1 and SARS-CoV-2 likely originated from bats before infecting humans[25,26], it is expected that novel coronaviruses or sarbecoviruses that bind to ACE2 will continue to spill over. Temman et al. recently demonstrated three bat sarbecoviruses specifically bound human ACE2, although there is currently no evidence of human infection[27]. Given these scenarios, future outbreaks are likely.

Whereas most vaccines are prophylactic, other approaches aimed at mitigating viral infection are administered post-infection. One approach is administration of antiviral or anti-inflammatory drugs, intended to interfere with processes critical to the viral life cycle; clinical antivirals include: Remdesivir, Paxlovid, and Molnupiravir[28]. There is evidence, however, that these antivirals may be losing their efficacy against new variants[29,30], with some immunocompromised patients already harboring Paxlovid-resistant variants[31]. Moreover, resistance to antivirals has been seen with other viruses such as human immunodeficiency virus (HIV)[32–34], human simplex virus (HSV)[35], and influenza A (IAV)[36]. Another therapeutic approach for patients with severe COVID-19 is the administration of anti-inflammatory corticosteroids. Dexamethasone, for example, has been shown to reduce mortality from COVID-19 by reducing the effects of cytokine storm[37]. However, since these anti-inflammatory drugs suppress the immune system

[1]Irell and Manella Graduate School of Biological Sciences, Beckman Research Institute, City of Hope, Duarte, CA, USA. [2]Department of Cancer Biology and Molecular Medicine, Beckman Research Institute, City of Hope, Duarte, CA, USA. [3]Division of Biology and Biological Engineering, California Institute of Technology, Pasadena, CA, USA. [4]These authors contributed equally: Juliet Gonzales, Tynan Young. ✉e-mail: jcwilliams@coh.org

and increase viral load, they are only useful to patients with the most severe cases of COVID-19[38]. Given the potential of antiviral resistance of SARS-CoV-2 and the risk associated with anti-inflammatory drugs, a therapeutic that overcomes these limitations is needed.

An alternative therapeutic approach to potentially inhibit all SARS-CoV-2 variants is a decoy strategy. As all SARS-CoV-2 variants require ACE2 interaction for infection, ACE2-based biologics offer a unique opportunity to be effective against all SARS-CoV-2 variants. Multiple groups have pursued an ACE2-centric approach, including the use of monomers[39], dimers[40,41], trimers[42–45], tetramers[46], pentamers and hexamers[47]. Some have focused on increasing the affinity of the ACE2 receptor to the spike protein through select mutations[48–55]; however, the inherent avidity of the interaction between endogenous ACE2 on target cells[56] and multiple copies of spike proteins per virus is expected to overcome a high-affinity ACE2 monomeric decoy. Moreover, these point mutations also exert a selective pressure for the virus and may lead to yet another escape variant.

To address avidity concerns, we[57] and others[48–55] have fused the ACE2 domain to an IgG Fc domain. Most of these efforts, however, have not incorporated structural information; namely, the trimeric nature of the spike protein and its C3 axis of symmetry, as observed in single-particle cryo-electron microscopy (cryo-EM) structures. Therefore, we sought trimeric scaffolds to fuse three ACE2 moieties and leverage the inherent symmetry of the spike trimer to maximize avidity. We have shown, using surface plasmon resonance (SPR), viral inhibition studies, and cryo-EM single-particle analysis, that this design potently inhibits each variant of SARS-CoV-2 tested as well as SARS-CoV-1 in vitro. Given its intended use in the clinic, we have also identified disulfide bonds in ACE2 and the scaffold that substantially increase the overall thermal and serum stability of the trimeric antagonist. Inhibition of enzymatic ACE2 activity was also incorporated. Collectively, the results of these efforts have afforded a potent, pan-SARS-CoV-2 antagonist and support its translation to the clinic, particularly to treat infected elderly and immunocompromised patients.

## Results

The overall affinity for a trivalent antagonist can be described as: $\Delta G_T = \Delta G_1 + \Delta G_2 + \Delta G_3 + \Delta G_S$, where $\Delta G_T$ is the total free energy and $\Delta G_S$ is the "connection" energy[58]. Using the relationship $\Delta G = -RT\ ln[K]$, this equation can be simplified to: $K_T = K_1\ K_2\ K_3 \cdot C_{eff}(R)$, where $C_{eff}(R)$ reflects the effective concentration of each additional binding moiety[59]. Of note, $C_{eff}(R)$ is sensitive to the geometry of the linker, accounting for entropy, sterics, etc. In the optimal case, the linker allows the individual ligands to match the geometry of the receptors, $C_{eff}(R) \sim 1$, and the binding constants are multiplicative. We and others routinely observed substantial gains in apparent affinity that approach multiplicative values[60,61]. As an ACE2 monomer binds the SARS-CoV-2 spike receptor binding domain (RBD) with an approximately $K_D \sim 10$–$50$ nM[52], it is theoretically possible to achieve an apparent binding affinity approaching $10^{-24}$ M.

To generate trimeric ACE2, the following characteristics were considered: immunogenicity risk, expression in mammalian cells, and stability. We first evaluated the geometry of ACE2 bound to the trimeric spike. Inspection of available cryo-EM models of SARS-CoV-2 spike protein bound to four ACE2 molecules (see "Methods") shows the C-termini of each ACE2 pointed away from the spike C3 symmetry axis and distant (113 ± 7 Å between Y613 residues)—compared to the N-termini of the ACE2, which are pointed toward the symmetry axis, closer (49 ± 5 Å between E22 residues), and sterically unimpeded (Fig. 1a). Thus, we determined a trimeric antagonist fused to the N-terminus of ACE2 would require a short peptide linker and would be less likely to form inter-spike interactions. Furthermore, we only considered trimerization domains of human origin, primarily to address potential immunogenicity concerns. Applying these constraints, four candidates stood out: pulmonary surfactant protein (SP-D)[62], Tumor Necrosis Factor-α (TNF-α)[63], collagen α1VIII[64], and collagen XVIII[65].

First, a Myc tag was fused to the C-terminal of the trimerization domains to determine whether the scaffold would support the expression of a trimeric antagonist. Expression was observed for both collagen trimerization domains whereas no protein expression was observed for either of the SP-D and TNF-α trimerization domains, confirmed by Western blot (SI Fig. 1). The collagen VIII trimerization domain is significantly larger than the collagen XVIII domain (44.7 kDa and 19.3 kDa, respectively), but more importantly, previous studies reported that collagen VIII melts at ~45 °C[66], whereas the collagen XVIII domain melts at ~85 °C (vide infra). Given these observations, the smaller, more thermally stable collagen XVIII trimerization domain was advanced to develop the trimeric antagonist.

Next, variable lengths of glycine-serine peptide linkers (0, 1, 3, 5, 18 amino acids) were tested to bridge the trimerization domain and ACE2 domains. To simplify the nomenclature, we refer to these constructs as TXA, where 'T' indicates the collagen XVIII trimerization domain, 'X' refers to the number of residues in the linker, and 'A' refers to ACE2 (Fig. 1a). Each TXA construct was expressed in Expi293 cells and purified as described in Methods. The trimeric ACE2 constructs eluted at the expected size, ~230 kDa, with high purity as determined by size-exclusion, high-performance liquid chromatography (SEC-HPLC) (SI Fig. 2).

The binding affinities of the ACE2 antagonists to the stabilized SARS-CoV-2 spike trimer were monitored using SPR. We find the trimeric ACE2 constructs consistently showed higher apparent binding affinities ($K_D$ values < 1 nM; Fig. 1b, Table 1) than monomeric ACE2 ($K_D$ of 49 ± 19 nM, consistent with reported values[43,45]. Figure 1b, Table 1), with similar on-rates ($k_a \sim 1.5 \times 10^5$ M$^{-1}$s$^{-1}$) but a magnitude lower off-rates ($k_d \sim 1 \times 10^{-4}$ s$^{-1}$ for ACE2 trimeric constructs; $k_d \sim 7 \times 10^{-3}$ s$^{-1}$ for ACE2 monomer). Similar SPR results were observed against the SARS-CoV-1 spike protein (Fig. 1c, Table 1). In these analyses, we fit the SPR data using a 1:1 interaction (e.g., one trimeric ACE2 binding to one spike trimer). Although we used a low-receptor-density SPR chip and we expect intramolecular interactions by design, we cannot rule out crosslinking distinct receptors (i.e., inter-spike interactions).

To determine neutralization potency, SARS-CoV-2 pseudotyped lentivirus expressing the spike protein from SARS-CoV-2 Wuhan-Hu-1 variant (Fig. 1d) was titrated with each trimeric spike as described previously[67]. Monomeric ACE2 inhibited with an IC$_{50}$ of 22 ± 5.5 nM, whereas the TXA constructs afforded IC$_{50}$ values < 1 nM. We also expressed a pseudotyped lentivirus expressing the Delta B.1.617.2 variant[68], Omicron BA.1 variant[69], and Arcturus XBB.1.16 variant (Fig. 1d) spike proteins[68]. Monomeric ACE2 exhibited an IC$_{50}$ of 24 ± 5.8 nM toward the Delta variant spike and an IC$_{50}$ of 16 ± 4.0 nM toward the Omicron variant, whereas all TXA constructs exhibited IC$_{50}$ values < 1 nM toward each of Delta, Omicron, and Arcturus variant spikes (Table 2). To test pan-strain viability, pseudotyped lentivirus expressing the SARS-CoV-1 (28 amino acid deletion—stabilized mutant) spike protein was titrated with monomeric ACE2 and trimeric ACE2 constructs. Monomeric ACE2 inhibited with two orders of magnitude higher IC$_{50}$ to SARS-CoV-1 spike (3200 ± 900 nM) than it did toward any of the SARS-CoV-2 variants; yet the TXA constructs afforded similar IC$_{50}$ values between strains (~0.6–2.2 nM) (Fig. 1e, Table 2).

Next, we turned to cryo-EM single-particle analysis with trimeric antagonists bound to SARS-CoV-2 spike trimer to validate the design (Fig. 2, Table 3). Starting with the 3-amino acid linker construct (T3A), we observed electron density for the spike protein, each of the three ACE2 moieties, and collagen XVIII trimerization domain of the T3A. The electron density between the trimerization domain and ACE2 moieties was discontinuous, reflective of the flexibility of the 3-amino acid linker. Moreover, modeling of the trimerization domain presents the "axis" as slightly 'tilted' (nonparallel) with respect to the C3 symmetry axis of the spike protein. Analysis of the other complexes, both with longer (5 and 18 amino acids) and shorter ("zero" amino acids) linker lengths, was conducted to better understand the role of the linker with respect to the symmetry of the trimeric antagonist. Notably, we did not observe discernible density for the collagen XVIII trimerization domains of the longer linker length constructs (T5A or T18A), whereas we did observe clear density connecting the trimerization

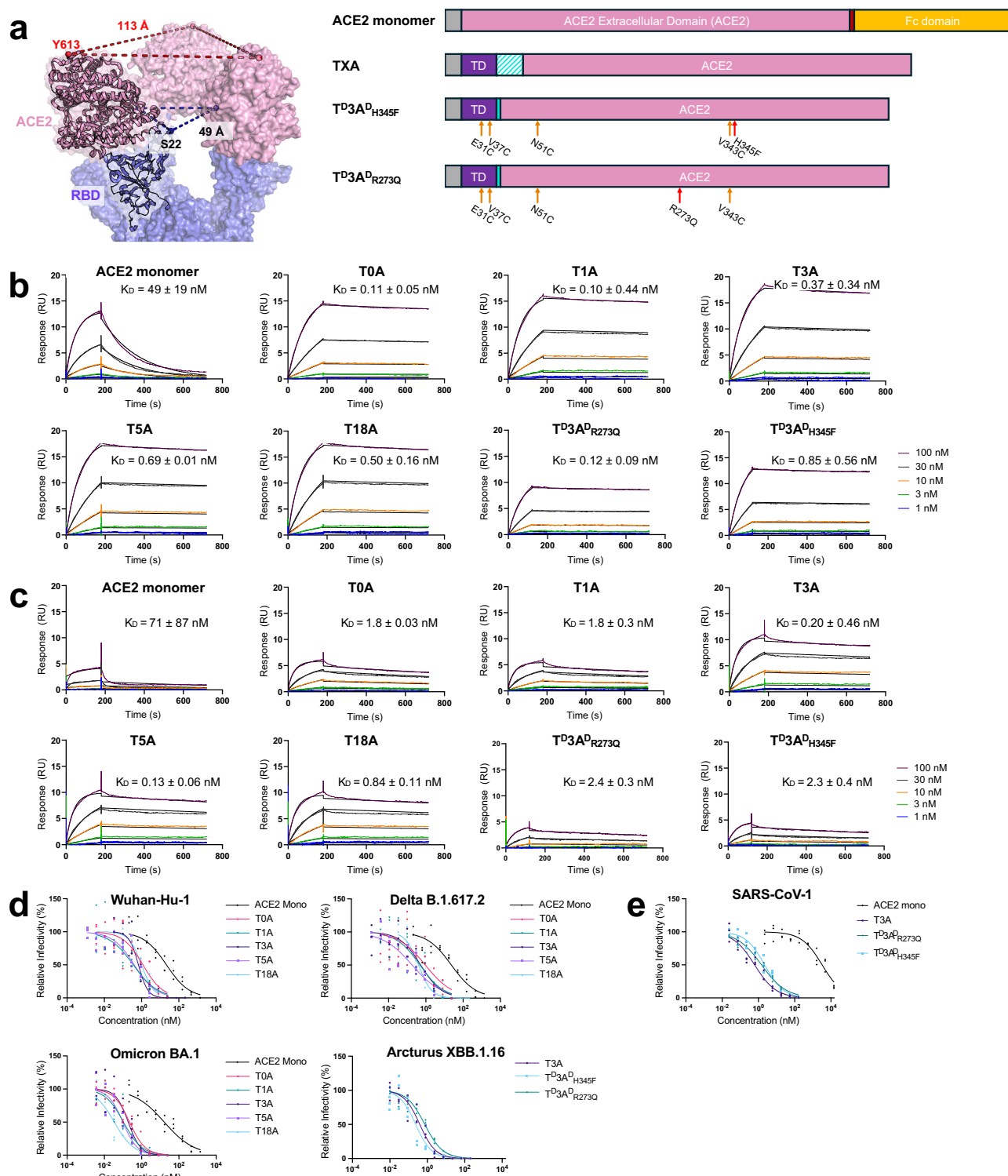

**Fig. 1 | Design principle, spike binding kinetics, and viral inhibition using trimeric ACE2 constructs. a** Diagram depicting density modeling for three ACE2 monomers (pink) binding the RBDs of SARS-CoV-2 spike (blue) with average distances between N-termini (residue S22) of five EM structures was 49 ± 5 Å (blue dashes), and between C-termini (residue Y613) was 113 ± 7 Å (red dashes); and construction of monomeric and trimeric ACE2 antagonists where trimeric antagonists fuse collagen XVIII trimerization domain (TD) to the N-terminus of ACE2 extracellular domain (ACE2). White-cyan dashes represent variable linker lengths. **b**, **c** SPR sensograms of ACE2 constructs binding to **b** SARS-CoV-2

recombinant spike HexaPro protein, and **c** SARS-CoV recombinant spike protein, *n* = 2. **d**, **e** ACE2 constructs inhibition in a single-round infection of HEK293-hACE2 cells against an HIV-based pseudotyped virus using full-length SARS-CoV-2 spike protein from: Wuhan-Hu-1, Delta B.1.617.2, or Omicron BA.1 strains, and select T3A constructs tested against Arcturus XBB.1.1 strain, *n* = 3; and **e** select T3A constructs and ACE2 monomer against stabilized, wildtype SARS-CoV-1 spike protein, *n* = 2. Errors described by standard deviation from *n* biological replicates. For clarity, all sensograms and infectivity curves generated from a single, representative experiment.

domain and ACE2 in the zero-linker length, T0A. Further, although the axis of symmetry through the modeled trimerization domain of T0A was nearly parallel to the spike C3 symmetry axis, it was off-center (SI Fig. 3).

We then attempted to evaluate the 'symmetry of interaction' between our ACE2 constructs with spike protein (T0A-S, T3A-S, T5A-S, and T18A-S; where S is the SARS-CoV-2 spike HexaPro protein) by likening the C3 symmetry to an equilateral triangle with all sides being equal length and a corresponding standard deviation between these lengths of zero (s.d. = 0). Thus, we calculated lengths (distances) and corresponding standard deviations between the Cα atoms of equivalent residues in the RBDs of the spike protein (residue D428; as determined by Fan et al.[70]) and in each of the ACE2 domains (residue N137; arbitrarily determined residue on ACE2) (Fig. 2, red and black dashed lines, respectively). The calculated mean distances and s.d. between spike RBD were: 41 ± 3.6 Å for T0A-S; 40 ± 1.0 Å for T3A-S; 40 ± 1.0 Å for T5A-S; and 42 ± 3.2 Å for T18A-S. Equating lower s.d. values to symmetry, we determine T3A and T5A to have higher symmetry than T0A or T18A. Next, we calculated mean distance and s.d. between ACE2: 149 ± 13.2 Å for T0A-S; 141 ± 7.6 Å for T3A-S; 144 ± 11.2 Å for T5A-S; and 142 ± 10.8 Å for T18A-S. Again, equating lower s.d. values to

symmetry, we now determine T3A to have the highest symmetry of binding between constructs. These distances and s.d. were compared to those of five cryo-EM structures of spike trimer with three ACE2 monomers bound (see "Methods") to compare symmetry and determine whether binding of our constructs potentially forces the RBD domains into un-natural or energetically unfavorable conformations. We observed average mean inter-RBD distances of 41 ± 1.9 Å and mean inter-ACE2 distances of 144 ± 6.0 Å. As expected, the s.d. values of inter-RBD and inter-ACE2 of these models without a trimeric domain are similar or lower than any of our complexes, indicating higher order symmetry. Regardless, the mean distances between spike RBD or ACE2 in each of our constructs fall within one standard deviation of that calculated from published structures and suggest that TXA constructs do not require or induce energetically unfavorable spike RBD conformations for interactions. Combined with clear density for the collagen XVIII trimerization domain and 'optimal' positioning of individual ACE2 domains, the T3A represented the 'best' compromise and was selected for further development.

Next, we turned our attention to determining the thermal stability of our constructs, as is frequently used to assess clinical candidates. Thus, we utilized differential scanning fluorimetry (DSF) using SYPRO Orange dye to measure melting points, starting with monomeric ACE2 ($T_m$ = 50.7 ± 0.2 °C) (Fig. 3a). To improve the stability of ACE2, we sought residue pairs capable of forming an engineered disulfide bond connecting residues distant in sequence (>50 residues) without interfering with RBD binding[65]. We chose three distinct pairs: N51–V343, A153–N277, and G399–T517. Of these, the disulfide bond, N51C–V343C (Fig. 3b), which bridges the N-terminal "end" with the "middle" region of ACE2, produced ACE2 proteins with the largest increase in melting temperature ($T_m$ = 55.1 ± 0.08 °C, Fig. 3a) and expression levels.

We then sought to prevent the dissociation of the collagen XVIII trimer via the introduction of intermolecular disulfide bonds, despite the high reported trimerization constant (where 50% of the trimer is monomeric) of 56 pM[49]. Cysteine mutations capable of creating inter-chain disulfide associations (i.e., C∘-C∘ distances less than 6 Å and absence of obvious steric clashes) were identified: G22–F'26, E31–V'37, and G(-1)–L'5 (SI Fig. 4). Expression of the individual collagen XVIII mutants and subsequent purification afforded low milligram quantities of protein. The parental construct yielded ~1.7 mg·L⁻¹ purified protein ($n = 1$); the disulfide mutants, E31C–V'37C and G(-1)C–L'5C constructs, yielded each ~1 mg·L⁻¹ protein each ($n = 1$); whereas the G22–F'26C mutant only yielded ~0.05–0.2 mg·L⁻¹ protein ($n = 2$). The formation of inter-chain disulfide bonds was confirmed by X-ray diffraction studies (Table 4). High-resolution crystal structures of the E31C–V'37C and G(-1)C–L'5C mutants afforded clear electron density for the disulfide bonds in both mutants (Fig. 3c, d, respectively). Comparison of these structures with the "parental" collagen XVIII trimerization domain (PDB ID: 3HSH) indicated little perturbation in the overall structure (root-mean-squared deviation (RMSD) = 0.71 Å and 0.52 Å for E31C–V'37C and G(-1) C–L'5, respectively).

**Table 1 | SPR binding affinities of ACE2 constructs to SARS-CoV-2 and SARS-CoV spike**

| SPR binding affinities of ACE2 constructs to SARS-CoV-2 Spike HexaPro | | | |
|---|---|---|---|
| Construct | $k_a$ (1/M·s) × 10⁵ | $k_d$ (1/s) × 10⁻⁵ | $K_D$ (M) × 10⁻⁹ |
| ACE2 monomer | 1.5 ± 0.02 | 730 ± 290 | 49 ± 19 |
| T0A | 1.5 ± 0.08 | 16 ± 6.0 | 0.11 ± 0.05 |
| T1A | 1.5 ± 0.5 | 1.5 ± 6.0 | 0.10 ± 0.44 |
| T3A | 2.1 ± 0.6 | 5.6 ± 4.6 | 0.37 ± 0.34 |
| T5A | 1.5 ± 0.09 | 11 ± 0.5 | 0.69 ± 0.01 |
| T18A | 1.7 ± 0.2 | 8.3 ± 1.7 | 0.50 ± 0.16 |
| $T^D3A^D_{H345F}$ | 1.5 ± 0.06 | 1.2 ± 0.8 | 0.85 ± 0.56 |
| $T^D3A^D_{R273Q}$ | 1.5 ± 0.1 | 1.7 ± 1.0 | 0.12 ± 0.09 |
| SPR binding affinities of ACE2 constructs to SARS-CoV Spike | | | |
| Construct | $k_a$ (1/M·s) × 10⁵ | $k_d$ (1/s) × 10⁻⁵ | $K_D$ (M) × 10⁻⁹ |
| ACE2 monomer | 1.7 ± 0.8 | 160 ± 200 | 71 ± 87 |
| T0A | 2.7 ± 0.3 | 48 ± 7.0 | 1.8 ± 0.03 |
| T1A | 2.6 ± 0.2 | 47 ± 3.0 | 1.8 ± 0.3 |
| T3A | 2.5 ± 1.6 | 3.7 ± 12.6 | 0.20 ± 0.46 |
| T5A | 2.2 ± 0.7 | 25 ± 3.0 | 0.13 ± 0.06 |
| T18A | 2.4 ± 0.4 | 20 ± 0.5 | 0.84 ± 0.11 |
| $T^D3A^D_{H345F}$ | 2.5 ± 0.3 | 61 ± 0.2 | 2.4 ± 0.3 |
| $T^D3A^D_{R273Q}$ | 2.3 ± 0.2 | 52 ± 6.0 | 2.3 ± 0.4 |

**Table 2 | Inhibition values of ACE2 construct against pseudotyped SARS-CoV-2 variants and SARS-CoV-1**

| Pseudotyped SARS-CoV-2 inhibition | | | | | | | | |
|---|---|---|---|---|---|---|---|---|
| Variant | IC₅₀ (nM) | | | | | | | |
| | ACE2 monomer | T0A | T1A | T3A | T5A | T18A | $T^D3A^D_{R273Q}$ | $T^D3A^D_{H345F}$ |
| Wuhan-Hu-1 | 22 ± 5.5 | 0.6 ± 0.2 | 0.2 ± 0.1 | 0.7 ± 0.1 | 0.4 ± 0.1 | 0.1 ± 0.2 | - | - |
| Delta B.1.617.2 | 24 ± 5.8 | 0.8 ± 0.4 | 0.4 ± 0.2 | 0.5 ± 0.2 | 0.2 ± 0.1 | 0.4 ± 0.1 | - | - |
| Omicron BA.1 | 16 ± 4.0 | 0.2 ± 0.1 | 0.1 ± 0.02 | 0.2 ± 0.04 | 0.1 ± 0.02 | 0.03 ± 0.01 | - | - |
| Arcturus XBB.1.16 | - | - | - | 0.4 ± 0.1 | - | - | 0.6 ± 0.2 | 0.4 ± 0.7 |
| Pseudotyped SARS-CoV inhibition | | | | | | | | |
| | IC₅₀ (nM) | | | | | | | |
| | ACE2 monomer | T0A | T1A | T3A | T5A | T18A | $T^D3A^D_{R273Q}$ | $T^D3A^D_{H345F}$ |
| SARS-CoV-1 (28aa del) | 3300 ± 900 | - | - | 0.6 ± 0.2 | - | - | 1.5 ± 0.4 | 2.2 ± 0.5 |

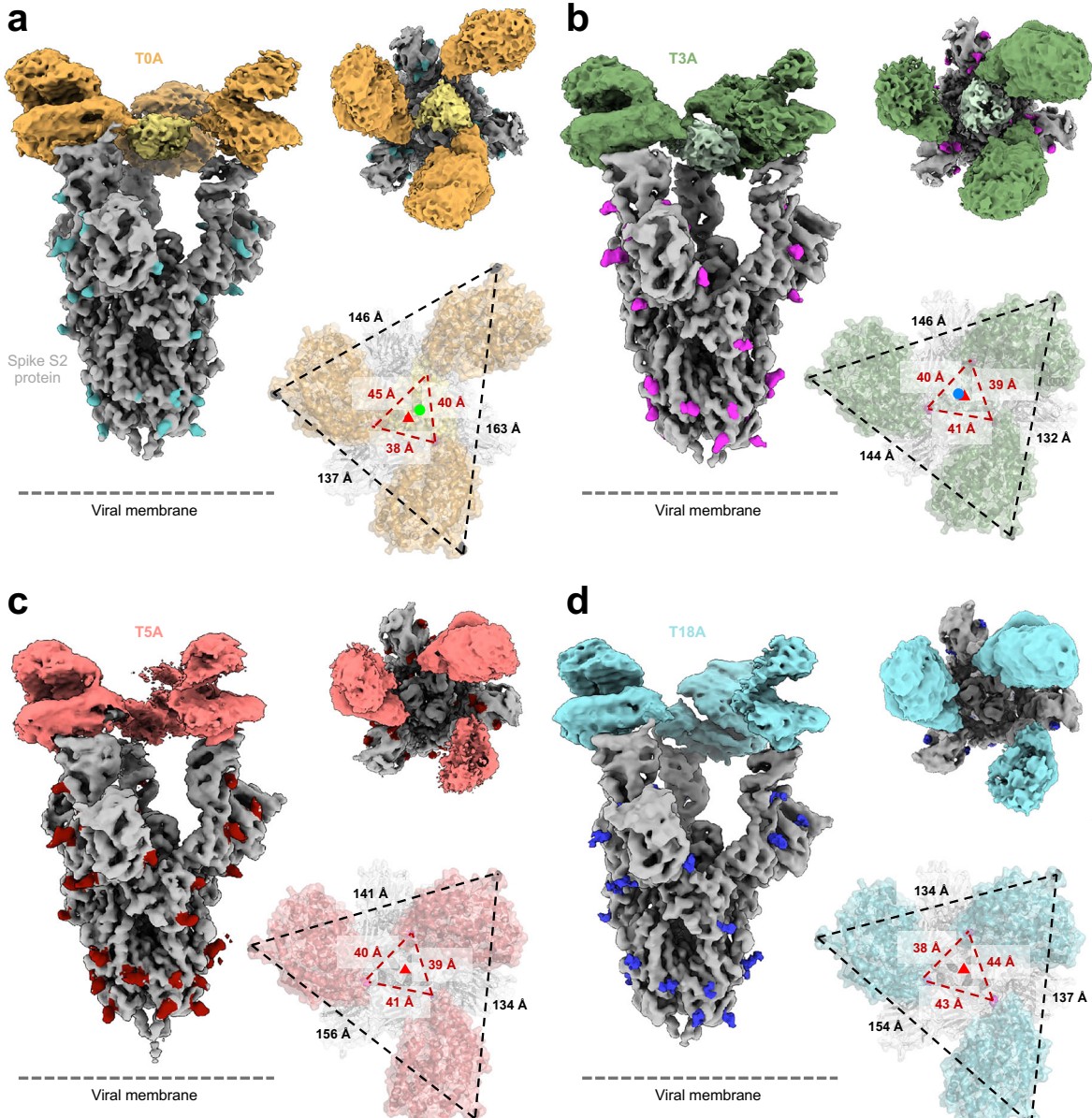

**Fig. 2 | Cryo-EM single-particle analysis electron densities and modeling of spike-bound trimeric ACE2 constructs. a–d** Electron density maps of spike HexaPro protein with bound trimeric ACE2 constructs: **a** T0A-S, **b** T3A-S, **c** T5A-S, and **d** T18A-S. Spike protein colored gray with glycans highlighted. ACE2 moieties of trimeric antagonists shown in orange, green, salmon, and teal, respectively. Each density shown perpendicular to the C3 axis of symmetry of the spike protein and looking down the axis of symmetry of the spike protein. Resolved density for collagen XVIII trimerization domains of **a** T0A and **b** T3A depicted in yellow and light green, respectively. Measurements in black depict distances between N137 Cα atoms of each ACE2 moiety, and measurements in red depict distances between D428 Cα atoms. The red triangles represent the C3 axis of symmetry through the spike protein. The C3 axis of symmetry for the modeled collagen XVIII trimerization domain is represented by green and blue circles in **a** T0A and **b** T3A, respectively. Density map images generated with ChimeraX. Axes of symmetry and model images generated with PyMOL.

Furthermore, the thermal stability of each collagen disulfide mutant was assessed by DSF. The 'wildtype' collagen XVIII trimerization domain produced a calculated melting temperature of 84.9 ± 0.2 °C (Fig. 3e, SI Fig. 5). Ascertaining the melting temperature of disulfide mutants proved more challenging. Given fluorescence intensity is proportional to "unfolded" protein concentration (0.1–1.0 mg·mL$^{-1}$), we expect similar concentrations of parental and mutant proteins (at ~0.5 mg·mL$^{-1}$) to produce similar intensity signals, albeit at different melting temperatures. However, two of the disulfide mutants (G(-1)C–L'5C and E31C–V'37C) either produced very low or no fluorescent intensity up to 100 °C. Considering the high melting temperature of the "parental" collagen XVIII, we posit that the melting temperatures of the disulfide mutants exceed 100 °C. Given the poor expression of the G22C–F'26C mutant with our trimeric ACE2 constructs, we proceeded with E31C–V'37C.

We next assessed the melting temperature of the trimeric ACE2 constructs with disulfide bonds. "Unmutated" T3A has a melting temperature of 47.6 ± 0.09 °C (Fig. 3f). The introduction of N51C–V343C T3A (termed T3A$^D$) increased the melting temperature to 55.9 ± 0.06 °C (Fig. 3f). The introduction of E31C–V'37C to the trimerization domain of T3A (termed T$^D$3A) afforded a melting temperature of 51.1 ± 0.03 °C (Fig. 3f). Combining the disulfide mutations, T$^D$3A with T3A$^D$ (termed T$^D$3A$^D$), further increased the melting temperature to 56.5 ± 0.01 °C (Fig. 3f).

As ACE2 catalyzes the critical function of converting angiotensin II to angiotensin (1–7), we[71] and others[52,72] sought to inhibit the enzymatic

**Table 3 | Cryo-EM data collection, refinement and validation statistics**

| | Trimeric ACE2 antagonist T0A in complex with SARS-CoV-2 spike HexaPro protein (EMDB-44724) (PDB 9BND) | Trimeric ACE2 antagonist T3A in complex with SARS-CoV-2 spike HexaPro protein (EMDB-44725) (PDB 9BNE) | Trimeric ACE2 antagonist T5A in complex with SARS-CoV-2 spike HexaPro protein (EMDB-44726) (PDB 9BNF) | Trimeric ACE2 antagonist T18A in complex with SARS-CoV-2 spike HexaPro protein (EMDB-44727) (PDB 9BNG) |
|---|---|---|---|---|
| **Data collection and processing** | | | | |
| Magnification | 105,000 | 105,000 | 105,000 | 45,000 |
| Voltage (kV) | 300 | 300 | 300 | 200 |
| Electron exposure (e–/Å²) | 60 | 60 | 60 | 60 |
| Defocus range (μm) | 0.5–3.0 | 0.5–3.0 | 0.5–3.0 | 0.5–3.0 |
| Pixel size (Å) | 0.832 | 0.832 | 0.832 | 0.869 |
| Symmetry imposed | C1 | C1 | C1 | C1 |
| Initial particle images (no.) | 1,593,457 | 1,616,068 | 1,333,041 | 2,795,476 |
| Final particle images (no.) | 194,801 | 225,636 | 220,795 | 534,324 |
| Map resolution (Å) | 3.19 | 3.43 | 3.33 | 3.73 |
| FSC threshold | 0.143 | 0.143 | 0.143 | 0.143 |
| Map resolution range (Å) | 2.8–3.5 | 3.1–3.8 | 3.2–3.5 | 3.5–4.2 |
| **Refinement** | | | | |
| Initial model used (PDB code) | PDB 6VXX, 6M0J | PDB 6VXX, 6M0J | PDB 6VXX, 6M0J | PDB 6VXX, 6M0J |
| Model resolution (Å) | 3.19 | 3.41 | 3.31 | 3.73 |
| FSC threshold | 0.143 | 0.143 | 0.143 | 0.143 |
| Model resolution range (Å) | 2.8–3.5 | 3.1–3.8 | 3.2–3.5 | 3.5–4.2 |
| Map sharpening $B$ factor (Å²) | 6.38 | 6.86 | 6.66 | 7.46 |
| **Model composition** | | | | |
| Non-hydrogen atoms | 40,266 | 40,155 | 38,931 | 38,779 |
| Protein residues | 4974 | 4966 | 4810 | 4808 |
| Ligands | 57 | 57 | 57 | 52 |
| **$B$ factors (Å²)** | | | | |
| Protein | 549.5 | 378.6 | 287.0 | 502.9 |
| Ligand | 218.2 | 222.3 | 198.0 | 217.4 |
| **R.m.s. deviations** | | | | |
| Bond lengths (Å) | 0.003 | 0.004 | 0.005 | 0.003 |
| Bond angles (°) | 0.540 | 0.546 | 0.955 | 0.502 |
| **Validation** | | | | |
| MolProbity score | 1.9 | 2.5 | 1.9 | 2.3 |
| Clashscore | 12.3 | 17.0 | 14.0 | 15.7 |
| Poor rotamers (%) | 1.3 | 4.0 | 0.4 | 2.8 |
| **Ramachandran plot** | | | | |
| Favored (%) | 96.4 | 96.0 | 95.9 | 96.5 |
| Allowed (%) | 3.6 | 4.0 | 4.1 | 3.5 |
| Disallowed (%) | 0 | 0 | 0 | 0.0 |

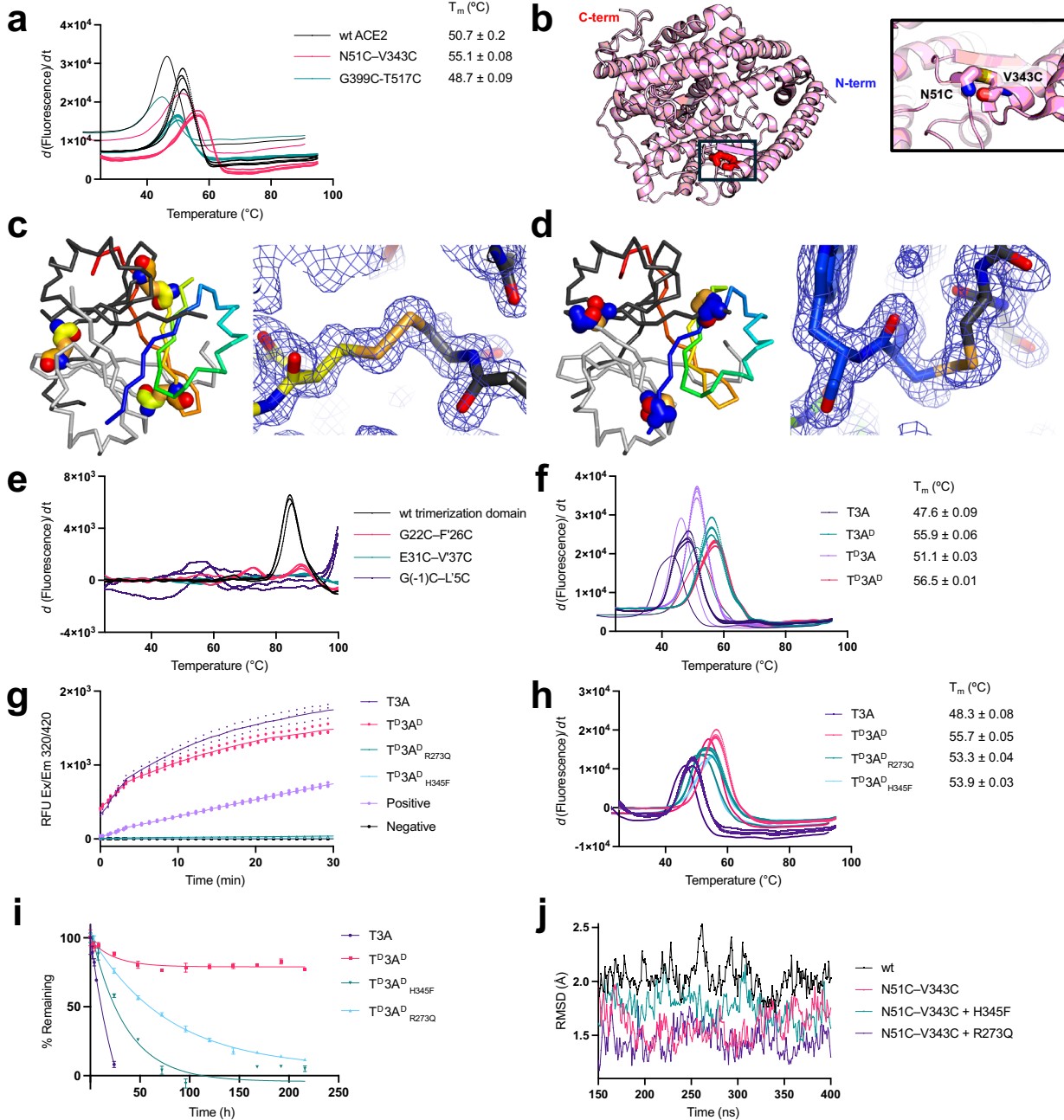

**Fig. 3 | Stabilization and inactivation of the enzymatic activity of T3A. a** Melting temperatures measured via DSF for wildtype monomeric ACE2 (wt, black line) and monomeric ACE2 with N51C–V343C or G339C-T517C (magenta and teal, respectively), $n = 2$. **b** A representative cartoon model of monomeric ACE2 (PDB ID: 6M0J) with introduced N51C–V343C disulfide bond highlighted (red sticks); model is colored pink with terminals labeled blue (N-) and red (C-). **c, d** Ribbon diagram and X-ray diffraction *2Fo-Fc* electron density maps at 1.0 s highlighting the introduced disulfide bond: **c** E31C–V'37C or **d** G(-1)C–L'5C; carbons are colored from N-(blue) to C-terminus (red) for a single chain, and gray for the other two chains. **e, f** Melting temperatures measured via DSF of **e** wt collagen XVIII trimerization domain (black line) and disulfide mutants: G22C–F'26C, E31C–V'37C, G(-1) C–L'5C (magenta, teal, purple, respectively), $n = 2$, and **f** unmutated T3A and disulfide mutants: T3A$^D$ (N51C–V343C in ACE2), T$^D$3A (E31C–V'37C in

trimerization domain), T$^D$3A$^D$ (E31C–V'37C in trimerization domain and N51C–V343C in ACE2), $n = 2$. **g** Enzymatic activity measured via cleavage of fluorescence peptide substrate analog with trendline depicting mean RFU, $n = 2$, and **h** melting temperature measured via DSF for T3A, T$^D$3A$^D$, T$^D$3A$^D_{R273Q}$, and T$^D$3A$^D_{H345F}$, $n = 3$. **i** T3A constructs incubated in rat serum at 37 °C were sampled over 216 h via ELISA using a SARS-CoV-2 spike HexaPro protein-coated plate. An HRP-conjugated anti-Myc tag was used to determine binding to spike protein with trendline fit to a single phase decay, $n = 4$. **j** MD simulations depicting RMSD (in Å) between 150–400 ns for wt ACE2 (black) and ACE2 with mutations: N51C–V343C (magenta); N51C–V343C with R273Q (purple); and N51C–V343C with H345F (teal), $n = 1$. Errors described by standard deviation from $n$ biological replicates. All melting, activity, and stability curves generated from a single, representative experiment for clarity.

activity in our therapeutic molecule to mitigate potential interference with the renin-angiotensin-aldosterone system[73]. Based on the ACE2 structure, H345 was identified as a residue involved in binding its substrate[72]. Thus, a H345F mutation in the ACE2 domains of T$^D$3A$^D$, termed T$^D$3A$^D_{H345F}$, was expressed and purified, and characterized using an ACE2 enzymatic activity

assay. T3A and T$^D$3A$^D$ displayed robust catalytic activity, whereas T$^D$3A$^D_{H345F}$ showed no enzymatic activity (Fig. 3g). Binding kinetics of T$^D$3A$^D_{H345F}$ and the "unmutated" T3A against SARS-CoV-2 spike and SARS-CoV-1 spike proteins remained indistinguishable (Fig. 1b, c, Tables 1, 2). The IC$_{50}$ of T$^D$3A$^D_{H345F}$ against pseudotyped SARS-CoV-2

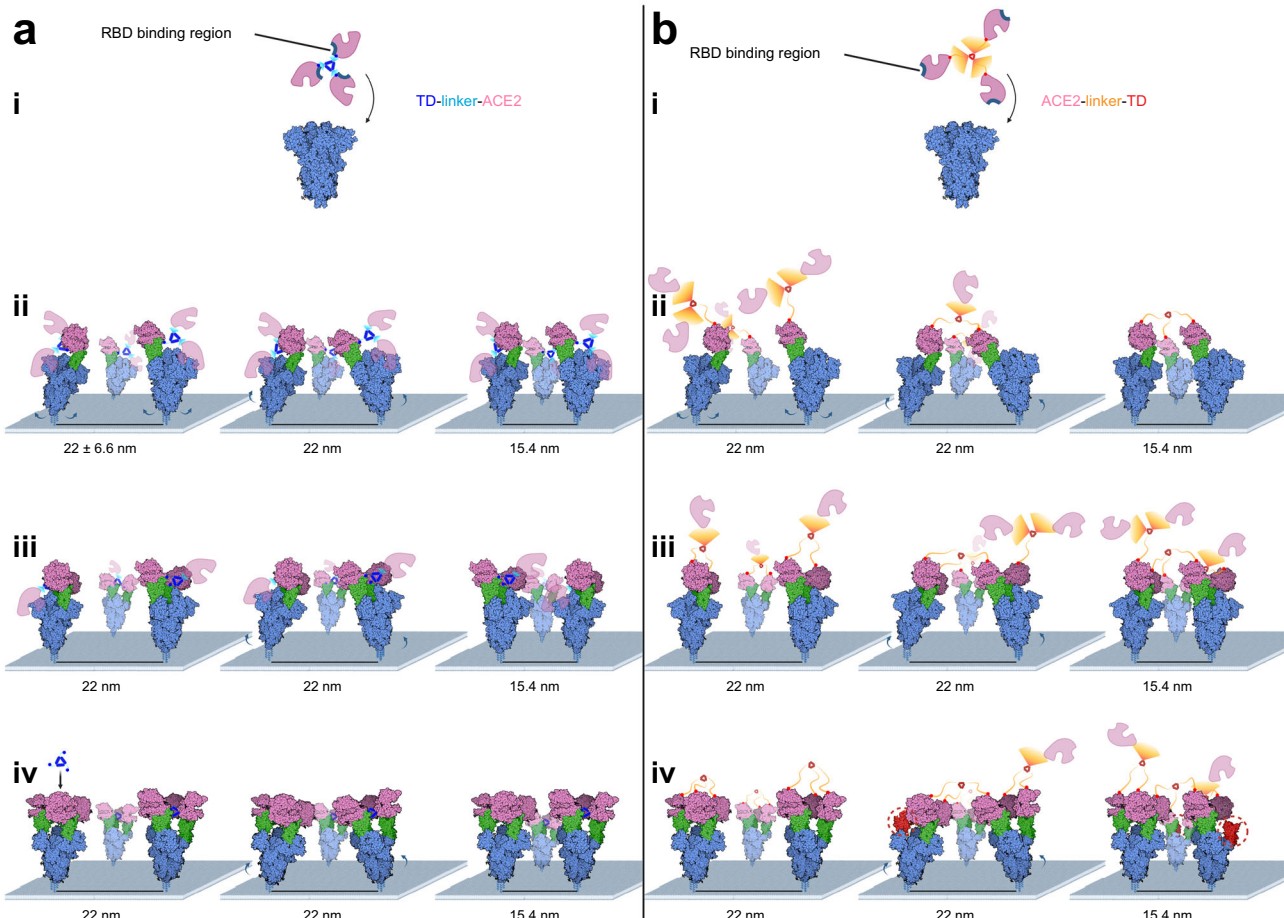

**Fig. 4 | Spike binding comparison of differentially oriented trimeric ACE2 antagonists.** (i) Cartoon representations of SARS-CoV-2 spike protein (blue) and trimeric antagonists. **a, b** Construct protein expression for **a** trimerization domain (blue open triangle) and linker (light blue curves/cones) fused to the N-terminus of ACE2 extracellular domain (ACE2; pink cartoon with RBD binding site highlighted teal), or **b** ACE2 fused via C-terminal linker (orange curves/cones) to trimerization domain (red open triangle). (ii–iv) Simulated binding of three equivalents of trimeric ACE2 antagonists to three spike proteins with each having (ii) one, (iii) two, or (iv) three RBD domains in the 'open'/ACE2-binding conformation. ACE2-bound RBDs are highlighted in green, and in red with circles when not bound/exposed. Unbound ACE2 monomers are depicted with pink cartoons and cones demonstrating the flexibility of linker domains. The N- and C-termini of bound ACE2 monomers are shown as red and blue circles, respectively. Created with Biorender.com.

Arcturus variant XBB.1.16 was also characterized, affording similar inhibition values to "unmutated" T3A (Fig. 1d, Table 2). Finally, DSF studies of $T^D3A^D_{H345F}$ indicated the phenylalanine substitution reduced thermal stability to $53.9 \pm 0.03$ °C (a decrease of 1.8 °C compared to $T^D3A^D$) (Fig. 3h).

Finally, serum stability assays were utilized to understand the impact of the disulfide and inactivation substitutions. A Myc tag was added to each of T3A, $T^D3A^D$, and $T^D3A^D_{H345F}$ antagonists. Similar melting temperatures determined by DSF indicate similar stability between antagonists with and without a Myc tag (SI Fig. 6). Each of these three constructs was incubated in rat serum at 37 °C over 216 h; serum aliquots were taken at ~24-h time points, frozen and stored at −80 °C until the time of analysis. These aliquots were diluted 20-fold with PBS buffer prior to addition to a 96-well plate coated with the spike HexaPro trimer, and the amount of ACE2 antagonist bound at each time point was determined quantitatively via anti-Myc-HRP luminescence signal. A rapid loss of signal after approximately 12.5 h (50% signal remaining with respect to the starting concentration) was observed for the "unmutated" T3A. The stabilized T3A construct, $T^D3A^D$, however, persisted (50% remaining) beyond our 216-h experiment. Surprisingly, the $T^D3A^D$ with inactivation mutation, H345F, degraded more rapidly: ~50% of the antagonist remained at approximately 24 h (Fig. 3i).

The substantial decrease in serum stability of the H345F mutant—despite very little changes in the melting temperature by DSF—led us to test an alternate deactivation mutation. R273 in ACE2 interacts with the ACE2 substrate, and its mutation to glutamine has been reported to inhibit enzymatic activity[74]. An R273Q substitution was introduced to the $T^D3A^D$ construct; this construct was expressed, purified, and the enzymatic activity was characterized. As expected, the R273Q mutation in T3A (termed $T^D3A^D_{R273Q}$) ablated enzymatic activity (Fig. 3g). SPR binding kinetics and affinity of $T^D3A^D_{R273Q}$ and "unmutated" T3A for both SARS-CoV-2 spike and SARS-CoV-1 spike proteins were indistinguishable (Fig. 1b, c, Tables 1, 2). In addition, inhibition of the pseudotyped SARS-CoV-2 Arcturus variant (XBB.1.16) with $T^D3A^D_{R273Q}$ afforded similar inhibition values to "unmutated" T3A (Fig. 1d, Table 3). The melting temperature of $T^D3A^D_{R273Q}$ was assessed by DSF and exhibited a 2.4 °C decrease ($T_m = 53.3 \pm 0.04$) compared to $T^D3A^D$ (Fig. 3h). The $T^D3A^D_{R273Q}$ construct showed a 2.5-fold increase in stability in rat serum at 37 °C compared to the H345F mutant; approximately 50% persisted for over 61 h (Fig. 3i, SI Fig. 7).

To further understand how each of these inactivation mutations (in combination with the N51C–V343C disulfide) affected the stability of the ACE2 molecule, we performed all-atom molecular dynamics (MD) simulations with: (1) 'wildtype' ACE2 (PDB ID: 6M0J); (2) N51C–V343C disulfide mutant; (3) N51C–V343C disulfide and H345F inactivation mutant; and (4) N51C–V343C disulfide and R273Q inactivation mutant (see "Methods"). To assess stability, we analyzed the RMSD over the simulation time. Here, we calculated the simulated distance that each atom moves with respect to its starting position from the crystal structure. Greater distances

**Table 4 | X-ray diffraction data collection and refinement statistics**

| | Collagen XVIII trimerization domain with introduced disulfide, E31C-V'37 C (PBD 9BNB) | Collagen XVIII trimerization domain with introduced disulfide, G(-1)C-L'5 C (PDB 9BNC) |
|---|---|---|
| Wavelength | 0.97934 | 0.97856 |
| Resolution range | 32.01–1.4 (1.45–1.4) | 37.72–1.5 (1.554–1.5) |
| Space group | P 21 21 21 | P 21 21 21 |
| Unit cell | 38.36 58.078 67.268 90 90 90 | 41.475 42.156 84.464 90 90 90 |
| Total reflections | 338,648 (12,550) | 320,785 (32,081) |
| Unique reflections | 29,542 (2435) | 24,405 (2381) |
| Multiplicity | 11.5 (5.2) | 13.1 (13.5) |
| Completeness (%) | 97.39 (81.41) | 99.89 (99.79) |
| Mean I/sigma (I) | 26.71 (5.05) | 11.67 (1.86) |
| Wilson B-factor | 13.26 | 22.67 |
| R-merge | 0.05122 (0.215) | 0.1211 (0.9007) |
| R-meas | 0.05341 (0.2396) | 0.1262 (0.9356) |
| R-pim | 0.01488 (0.1018) | 0.03522 (0.2513) |
| CC1/2 | 0.999 (0.956) | 0.996 (0.862) |
| CC* | 1 (0.989) | 0.999 (0.962) |
| Reflections used in refinement | 29,539 (2435) | 24,390 (2381) |
| Reflections used for R-free | 1427 (126) | 1180 (119) |
| R-work | 0.1710 (0.1931) | 0.2015 (0.2732) |
| R-free | 0.1959 (0.2401) | 0.2268 (0.3095) |
| CC (work) | 0.965 (0.878) | 0.964 (0.868) |
| CC (free) | 0.960 (0.851) | 0.937 (0.708) |
| Number of non-hydrogen atoms | 1551 | 1441 |
| Macromolecules | 1364 | 1331 |
| Ligands | 22 | 0 |
| Solvent | 175 | 110 |
| Protein residues | 170 | 167 |
| RMS(bonds) | 0.010 | 0.012 |
| RMS(angles) | 1.10 | 1.26 |
| Ramachandran favored (%) | 95.12 | 96.25 |
| Ramachandran allowed (%) | 4.88 | 3.75 |
| Ramachandran outliers (%) | 0 | 0 |
| Rotamer outliers (%) | 0.72 | 0.71 |
| Clashscore | 2.55 | 7.55 |
| Average B-factor | 20.37 | 28.13 |
| Macromolecules | 18.70 | 27.39 |
| Ligands | 49.45 | - |
| Solvent | 31.38 | 37.06 |

indicate the molecule has more intermolecular "fluctuations" and is less stable. Indeed, we find less intermolecular fluctuations in each of the mutant models compared to the 'wildtype' model (Fig. 3j). The lowest average RMSD was observed in the N51C–V343C, R273Q mutant (avg = 1.46 Å), followed by the N51–V343C disulfide mutation (avg = 1.60 Å), then the N51C–V343C, H345F mutant (1.76 Å), and finally 'wildtype' ACE2

(2.04 Å). Taken together, these limited MD simulations are consistent with the serum stability studies: the disulfide bond in the ACE2 stabilizes the molecule, and the R273Q inactivation mutation shows reduced structural fluctuations compared to the H345F mutation, correlating with an increase in serum stability. These results are consistent with reports that show reduced RMSD values are associated with protein stability[75,76].

Collectively, the enzymatically inactivated construct, $T^D 3A^D_{R273Q}$, closely matches the C3 symmetry of the spike, binds to the spike protein with high affinity, demonstrates high viral inhibition potency against SARS-CoV-1 and all SARS-CoV-2 variants tested, and has high stability in serum. Thus, it is a viable development candidate to produce a universal therapeutic against SARS-CoV-2 and other ACE2-utilizing entry viruses.

## Discussion

While the pandemic has subsided, COVID-19 remains a major health concern. From January 2024 to May 2025, over half a million people have contracted the disease worldwide[2]. And while vaccines were developed in record time, saving numerous lives and significantly slowing down the disease, SARS-CoV-2 continues to evolve, rendering vaccines and therapeutics less effective. Undoubtedly, viral evolution is leading to the emergence of resistance to antivirals such as Paxlovid. Given this, we have taken an ACE2-centric antagonistic approach. The basis of this approach is simple: SARS-CoV-1 and all SARS-CoV-2 variants require ACE2 on the host cell to gain access and complete their viral life cycle, and by preferentially binding the antagonist over the host ACE2, the virus cannot enter and infect the host cell.

Decoy receptor antagonists using different scaffolds have been developed and have had varying degrees of success in antagonizing SARS-CoV-2 in organoids[77], animal models[44,48,51,78], and even human clinical trials[39]. Our approach differs from these in how we constructed the decoy antagonist. Namely, we understood that maximal affinity through avidity can be achieved by binding all three RBDs with an optimized trimeric linker (i.e., $C_{eff}(R)$ ~1). We also recognized that the N-termini of each of the ACE2 monomers (when bound to the spike trimer) were effectively pointing toward the spike symmetry axis, significantly close to one another and free of steric hindrance (Fig. 1a). In this order, trimerization-linker-ACE2, our antagonist differs from published dimeric and trimeric ACE2 decoys, which use an ACE2-linker-trimerization/dimerization domain orientation. As shown here, our approach allows for minimal amino acid linker lengths between ACE2 and trimerization domain, even direct fusion to the trimerization domain. As a result, once one of the ACE2 monomers of our trimeric antagonist binds a spike RBD, the other two are poised to immediately bind the second and third RBDs (Fig. 4aii–iv). In other words, this design is expected to cap all three RBDs within a spike trimer with extremely high affinity.

While our design seeks to maximize affinity through avidity, ultimately capping each RBD in an intra-spike manner, we cannot rule out the possibility that using an ACE2-linker-multimerization domain approach enables inter-spike crosslinking (i.e., between two adjacent spike trimers), which could be effective. With ACE2 inter-C-termini distances of ~11 nm, an ACE2-linker-multimerization antagonist would require sufficiently long linkers to bridge these termini, including avoiding inherent steric clashes. Considering average inter-spike distances of ~22 nm, inter-spike interactions could occur within as little as ~11 nm, the same distance as the required minimal linker. Thus, using an ACE2-linker-multimerization antagonist can, theoretically, bind via intra-spike interactions, or via inter-spike crosslinking (Fig. 4bii–iv). Inter-spike crosslinking, as opposed to intra-spike capping, may, therefore, result in free/unbound spike RBDs (Fig. 4b, iv). It remains to be determined if 100% of spike RBDs need to be blocked to inhibit host cell infection. In other words, if only one or two of the individual spike RBD domains are blocked, can one or more remaining RBDs interact with and infect the host cell? Clearly, these different scenarios require significantly more experimentation with alternative and more sensitive assays to address these questions.

Beyond the architecture of our trimer-linker-ACE2 constructs, we took additional steps to incorporate physical properties needed to advance this design into the clinic. First, we used a trimeric domain of human origin, intentionally selected to reduce or avoid eliciting an immune response. Second, we introduced disulfide bonds to increase thermal stability and reduce proteolysis; in the trimerization domain, these also served to prevent complex dissociation. Third, we inactivated the enzymatic activity. Surprisingly, the initial single point mutation, H345F, rendered the inactivated, stabilized trimeric antagonist more susceptible to serum activity than the enzymatically active, stabilized antagonist. While the serum stability was greatly improved with an alternative inactivating point mutation, R273Q, these findings suggest additional improvement is possible.

Our serum stability studies are clearly not the same as in vivo animal experiments; unfortunately, in vivo pharmacokinetic studies are not currently possible. However, these studies do highlight intrinsic differences between ACE2 decoys. Of note, the half-lives of reported ACE2 decoys are: monomeric ~10 h[79]; dimeric ~30–65 h[48,53,74], and trimeric ~1 h[44]. We note the trimeric construct fuses ACE2 to the *N-terminus* of a trimeric "foldon", a scaffold protein of bacteriophage origin, which has a trimerization constant of 5–30 mM[80] (compared to 56 pM for collagen XVIII trimerization domain). In other words, the foldon-based decoy is likely no longer trimeric at low concentrations, which might explain the short half-life. Thus, we anticipate a construct with a lower trimerization domain constant and human origin to have a longer half-life.

Our plan in the immediate future is to conduct animal studies using the optimized T3A antagonist across multiple SARS-CoV-2 variants. Although we have addressed many critical physical properties necessary for animal studies, we recognize our molecule may require additional modifications, such as the addition of an albumin-interacting domain to further increase serum stability. We will also compare this treatment with other formats where we have incorporated similar mutations to address stability (e.g., AXT with the appropriate disulfides and inactivation mutation).

Collectively, we have incorporated extensive structural information to generate a potent, symmetry-matched therapeutic that is expected to be active across all SARS-CoV-2 variants, including future SARS strains. Assuming favorable outcomes in animal studies, we suggest that this decoy antagonist will be especially useful for immunocompromised patients, including cancer patients undergoing stem cell implantation.

## Methods

### Structural analysis

The following PDBs were used for distance and symmetry analysis of the ACE2 binding spike protein: 7CT5, 7KJ4, 7KMS, and 7KNI (7A98 was only used in spike symmetry analysis). Residues E22 and Y613 of each ACE2 were chosen to represent the N- and C-terminus, respectively. Residue D428 of each RBD on the spike protein and residue N137 of each ACE2 were used to represent spike and ACE2 symmetry, respectively. PyMOL v2.4.1 was used to determine distances between Cα atoms for all sets of residues for each PDB. Every distance was measured and averaged, and standard deviations were calculated in a spreadsheet ($n = 4$ models for all ACE2 analysis, $n = 5$ for spike symmetry analysis).

### Protein expression and purification

To generate ACE2 monomer, human extracellular ACE2 domain (residues 18–615) was fused with a C-terminal TEV cleavage site and Fc immunoglobulin domain to generate *ACE2TEVFc* and were produced in Expi293 cells (Thermo Fisher Scientific).

To generate TXA constructs, the trimerization domains were fused onto the N-terminus of the human extracellular ACE2 domain (residues 18–615) with various G/S linkers between. A secretion signal and 6× His tag were fused onto the N-terminus of the trimerization domain for expression and purification of the constructs. All mutations were added using a Q5 site-directed mutagenesis kit (New England Biolabs). All primers and gene-blocks were ordered through Integrated DNA Technologies (IDT).

All constructs were transfected into Expi293 cells by transient transfection using Expi293 expression kit (Thermo Fisher Scientific). Cells were incubated for 7 days at 37 °C with 5% $CO_2$.

For ACE2-TEV-Fc purification, Expi293 cells expressing the protein were spun down at 8000×$g$ for 45 min, and the supernatant was collected and filtered through a 0.45 and 0.22 μm membrane sequentially. The filtered supernatant was loaded onto a protein G resin FF pre-packed column (Genscript). The column was washed with PBS and eluted using 0.1 M acetic acid (pH 3.5), and the eluted protein was neutralized with a 1 M Tris-HCl (pH 9.0). To generate monomeric ACE2, the eluted protein was then cleaved using TEV protease[80] overnight at room temperature. To remove TEV-Fc from the cleaved ACE2 monomer, the protein was loaded onto protein G column (Genscript) and 5 mL Ni Sepharose (HisTrap Excel, Cytiva) sequentially. The unbound fraction from both columns was collected. The ACE2 monomer was then further purified via size-exclusion chromatography (Cytiva, Superdex 200 Increase 10/300 GL column).

For TXA purification, the cells were spun down at 8000 × $g$ for 45 min, the supernatant was collected and filtered through a 0.45 and 0.22 μm membrane sequentially. The filtered supernatant was loaded onto a 5 mL Ni Sepharose (HisTrap Excel, Cytiva), washed with a 10 mM imidazole and eluted with a 300 mM imidazole. The eluted protein was then concentrated and run on a size-exclusion chromatography (Cytiva, Superdex 200 Increase 10/300 GL column).

Expression and purification of the SARS-CoV-2 spike HexaPro protein (Addgene #154754) construct was previously described[81]. Transfected Expi293 cells expressing the protein were spun down at 8000×$g$ for 45 min, and the supernatant was collected and filtered through a 0.45 and 0.22 μm membrane sequentially. The filtered supernatant was loaded onto a 5 mL Ni Sepharose (HisTrap Excel, Cytiva), washed with a 10 mM imidazole and eluted with a 300 mM imidazole. The eluted protein was then concentrated and run on a size-exclusion chromatography (Cytiva, Superdex 200 Increase 10/300 GL column).

Recombinant SARS-CoV-1 spike protein plasmid 2P (residues 12–1193) (Genbank: AAP13441.1)[82]. Expression and purification were performed similarly to the SARS-CoV-2 spike HexaPro protein.

Purified proteins (SI Fig. 1) were subjected to size-exclusion chromatography-high-pressure liquid chromatography (SEC-HPLC) using a 1260 Infinity II system equipped with a TSKgel UP-SW aggregate column (Agilent) and run at 0.15 mL min⁻¹ using a buffer containing 100 mM $Na_2H_2PO_4$, 100 mM $Na_2SO_4$, 0.05% (w/v) azide, pH 6.7.

'Wildtype' collagen XVIII trimerization domain and disulfide mutants were generated by fusing a 6× His tag, cleavable smt3 protein, and A/S/G-containing linker to the N-terminus of the trimerization domain. The collagen XVIII trimerization domain gene was synthesized and cloned into a Kanamycin-resistant *pSMT3* (U0086DA120-3) vector by Genscript and subsequently transformed into BL21 (DE3) *E. coli* competent cells (New England Biolabs).

Cells were grown from a single, isolated colony into 100 mL LB/Kan⁺ with shaking at 37 °C and 200 rotations per minute (RPM) overnight and scaled up to 6 × 1 L cultures before inducing with 0.4 mM IPTG at $OD_{600}$ 0.8. Cells were collected after 4 h post-induction and pelleted at 4000 RPM for 15 min before being allowed to freeze at −80 °C overnight. Cells were resuspended in 50 mM Tris-HCl, pH 8.0, 300 mM NaCl, 10% glycerol, 0.4% Triton-X supplemented with ~0.1–1.0 mg·mL⁻¹ lysozyme, 50 μg·mL⁻¹ DNase, and 1 protease inhibitor tablet (MedChemExpress)/25 mL buffer and lysed via three passes of cell disruption at 15,000–18,000 psi using an Emulsiflex-C3 cell homogenizer. Cellular debris and membrane were pelleted via ultracentrifugation at 48,000 RPM for 45 min and supernatant/lysate was isolated.

The lysate was loaded onto a 5 mL Ni Sepharose (HisTrap Excel, Cytiva) column equilibrated in 50 mM Tris-HCl, pH 8.0, 300 mM NaCl, 5 mM imidazole. The column was washed with a 50 mM Tris-HCl, pH 8.0, 300 mM NaCl, 15 mM imidazole buffer before eluting with 50 mM Tris-HCl, pH 8.0, 300 mM NaCl, 300 mM imidazole. Eluted protein was dialyzed via 3k MWCO snakeskin dialysis tubing overnight at 4 °C into a 50 mM

Tris-HCl, 300 mM NaCl solution containing recombinant Ulp-1 protease to cleave the smt3 protein of the construct from the trimerization domain. The overnight cleaved protein was then loaded back over a cleaned, 5 mL HisTrap Excel column (pre-equilibrated in 50 mM Tris-HCl, 300 mM NaCl, 5 mM imidazole) and the flowthrough containing the cleaved trimerization domain was further purified by size exclusion on a Superdex 75 Increase 10/300 GL column (Cytiva) in 10 mM Tris-HCl, 50 mM NaCl. The peak fractions were collected and pooled for further use.

### Binding assay by SPR

Binding of ACE2 constructs to soluble, SARS-CoV-2 spike HexaPro protein or soluble SARS-CoV-1 spike trimer was measured via SPR. Each ACE2 construct was diluted using an HBS-EP$^+$ running buffer (10 mM HEPES pH 7.4, 150 mM NaCl, 3 mM EDTA, 0.05% [v/v] surfactant P20). Both soluble S HexaPro protein and soluble SARS-CoV-1 spike trimer were immobilized to a CM5 S-series sensor chip (GE Healthcare) using standard EDC/NHS amine coupling in 10 mM acetate buffer, pH 5.0, at a density suitable for kinetics, 100 response units (RU). After equilibrating in the running buffer for 7 min, ACE2 constructs at various concentrations (1–100 nM) were flowed over the sensors with immobilized spike protein at a rate of 30 µL min$^{-1}$ for 5 min to measure association rate ($k_a$). A control sensor without spike protein was used as reference. The chip was regenerated using a 100 mM glycine buffer (pH 2.0) followed by HBS-EP+. SPR studies were performed using a Biacore T200 SPR system (GE Healthcare), and data were processed using Biacore T200 Evaluation software v.3.0 using a 1:1 binding model. Experiments were repeated in duplicate with similar results.

### HIV-based pseudovirus assay

Pseudotyped virus particles were produced in HEK293T cells (BEI Resources) as previously described[67]. Briefly, HEK293T cells were seeded in a six-well plate ($5 \times 10^5$ per well) and transfected 24 h post-seeding with the following plasmids: 1 µg of lentiviral backbone (NR-52516); 0.22 µg each of plasmids *HDM-Hgpm2* (NR-52517), *pRC-CMV-Rev1b* (NR-52519), and *HDM-tat1b* (NR-52518); and 0.34 µg viral entry protein for: SARS-CoV-2 Spike Wuhan-Hu-1 (BEI Resources NR-53742), SARS-CoV-2 Delta strain (Addgene plasmid #155130), SARS-CoV-2 Omicron BA.1 (Addgene plasmid #180375), SARS-CoV-2 Arcturus XBB.1.16 (Addgene plasmid #201189), or SARS-CoV-1 WT (Addgene plasmid #170447). At 24 h post-transfection, the media was exchanged. At 60 h post-transfection, the supernatant containing the virus was collected, filtered through a 0.45 µm SFCA low protein-binding filter, aliquoted and stored at −80 °C until use.

Neutralization of HIV-based pseudovirus containing full-length spike proteins from: SARS-CoV-2 Wuhan-Hu-1, SARS-CoV-2 Delta strain, SARS-CoV-2 Omicron BA.1, SARS-CoV-2 Arcturus XBB.1.16, and SARS-CoV-1 WT were measured using a single-round infection assay in HEK-293T-hACE2 cells (obtained through BEI Resources, NIAID, NIH: Human Embryonic Kidney Cells (HEK-293T) Expressing Human Angiotensin-Converting Enzyme 2, HEK-293T-hACE2 Cell Line, NR-52511).

HEK293T and HEK-293T-hACE2 cells were incubated in Dulbecco's modified Eagle's medium (DMEM, GIBCO) supplemented with 10% heat-inactivated fetal bovine serum (FBS, GIBCO) and 100 U·mL$^{-1}$ penicillin, and 100 µg·mL$^{-1}$ streptomycin (Thermo) at 37 °C with 5% CO$_2$. For neutralization assays, 3-fold serial dilutions of the ACE2 constructs were performed in triplicates. Diluted treatment and pseudovirus were incubated for 1 h at 37 °C and then added to plates containing HEK-293T-hACE2 cells ($1.25 \times 10^4$ per well). Wells containing cells and media only were negative controls, and wells containing cells and pseudovirus without treatment were positive controls. Samples and cells were incubated for 48 h at 37 °C with 5% CO$_2$ results. Following incubation, excess medium was removed, and cells were lysed with BrightGlo luciferase reagent (Promega). Relative luminescence units (RLUs) were normalized to values from control cells. BioTek Synergy H1 (Agilent) multimode plate reader was used to read plates. Data was normalized to the RLU without treatment as 100%, and the RLU without pseudovirus as 0% for each variant. Half-maximal inhibitory concentrations (IC$_{50}$) values are reported in molar concentrations (to

account in ACE2 monomer versus TXA difference in molecular weight) and were calculated using non-linear regression (GraphPad Prism). Experiments were repeated in triplicate with similar results.

### Cryo-EM sample preparation and data acquisition

T0A bound to SARS-CoV-2 spike HexaPro protein (T0A-S complex), T3A bound to SARS-CoV-2 spike HexaPro protein (T3A-S complex), T5A bound to SARS-CoV-2 spike HexaPro protein (T5A-S complex), and T18A bound to SARS-CoV-2 spike HexaPro protein (T18A-S complex) were each diluted in PBS buffer to 0.27 mg·mL$^{-1}$, ~0.33 mg·mL$^{-1}$, 0.30 mg·mL$^{-1}$, and 0.31 mg·mL$^{-1}$, respectively. A three µL aliquot of each complex was applied to cryo-EM grids (Quantifoil R 1.2/1.3 Cu, 300-mesh or Protochips CF-1.2/1.3-3Cu-50, 300 mesh) that had been glow-discharged for 1 min at 20 mA with a 30 s hold using a PELCO easiGLOW (Ted Pella) and plunge-frozen in liquid ethane using a Vitrobot Mark IV (FEI) at 4 °C and 100% humidity with a blot time of 4.0 s and blot force of either 0 or −2.

Data for the T0A-S, T3A-S, and T5A-S complexes were collected on a 300 keV Titan Krios electron (FEI) microscope using a Gatan Energy Filter and Gatan K3 Summit direct detection camera in super-resolution mode at the Cryo-EM Center–Beckman Institute at the California Institute of Technology (Caltech, Pasadena, CA). The automated software SerialEM[83] was used to capture movies in counting mode at a ×105,000 magnification and corresponding pixel spacing of 0.4275 Å. A dose rate of 18.80 pixels per second was used for T0A-S complex, 15.80 pixels per second for T3A-S complex, and 19.59 pixels per second for T5A-S complex to yield a final exposure of 60 e$^-$/Å$^2$. Movies were fractionated into 40 frames of 41.28 ms for a total exposure of 1.651 s per movie for T0A-S and T5A-S complexes, and 40 frames of 53.93 ms for a total exposure of 2.157 s per movie for T3A-S complex. A single session yielded 5116 micrographs for T0A-S, 5234 micrographs for T3A-S complex, and 4457 micrographs for T5A-S complex using a defocus range between 0.5 and 3.0 µm.

Two datasets for the T18A-S complex were collected on a 200 keV Talos Arctica electron (FEI) microscope using a Gatan K3 Summit direct detection camera in super-resolution mode at the Caltech Cryo-EM Center. The SerialEM automated software was used to capture movies in counting mode at a magnification of ×45,000 and corresponding pixel spacing of 0.4486 Å. Dose rates of 17.62 pixels per second and 17.67 pixels per second yielded final exposures of 60 e$^-$/Å$^2$. Movies were fractionated into 40 frames of 69.50 ms and 63.25 ms for total exposures of 2.780 s and 2.530 s per movie, respectively. Two sessions yielded 5123 and 2866 micrographs for a total of 7988 micrographs collected with a defocus range between 1.0 and 3.0 µm.

All datasets were motion corrected with patch motion correction using a crop factor of ½ with Patch CTF in cryoSPARC v3.3.2[84]. Particle picking was performed with Blob Picker in cryoSPARC using a particle diameter of 150–300 Å. Movies and particle picks were inspected and filtered before being extracted with a box size of 384 pixels. Particles were subjected to rounds of 2D classification and class selection in cryoSPARC, and the selected particles were used to create 5 ab initio volumes. Particles contained within nonsensical, junk volumes were selected against, and the remaining particles were initially refined using heterogeneous refinement and subsequently refined using non-uniform refinement in cryoSPARC. The resulting map was sharpened using Sharpen in cryoSPARC. Maps were evaluated and figures generated using UCSF ChimeraX[85,86] and exported for structural modeling and refinement.

### Cryo-EM structure modeling and refinement

An initial model of T3A-S complex was generated by removing the RBD domains of a docking a single-particle cryo-EM structure of SARS-CoV-2 spike HexaPro protein (PDB ID: 6VXX) and docking the resulting model into the cryo-EM electron density map in Phenix v1.21.2-5419[87] using phenix.dock_in_map. A crystal structure containing ACE2-bound RBD (PDB ID: 6M0J) was then manually placed into the electron densities using PyMOL[88] (version 3.0) for each of the three spike RBD domains, and the amalgamated structure was subjected to real-space, rigid-body refinement

in phenix.real space_refine. The rigid-body refined structure was submitted to multiple rounds of analysis and manual fitting in Coot v0.9.8.8[89] and real-space refinement in phenix.real_space_refine. The resulting T3A-S complex model was used as a reference in determining structures for T0A-S, T5A-S, and T18A-S complexes before modelling in the collagen XVIII trimerizations domain (PDB ID: 3HSH) for the T0A-S and T3A-S complexes. After placement of the trimerization domain, the T0A-S and T3A-S structures were subjected to final rounds of refinement using phenix.real_space_refine and model building in Coot. The final model was evaluated, and distances were assessed using PyMOL.

### Differential scanning fluorimetry (DSF)

DSF denaturation curves were used as a proxy for protein stability. DSF reactions (10 μL) consisted of 0.5 mg·mL$^{-1}$ protein and 5 × SYPRO™ Orange final (Life Technologies) in 1× PBS (pH 7.4). Reactions were carried out in triplicate in a 96-well plate on a QuantStudio 3 real-time PCR instrument (Applied Biosystems) following the manufacturer's protocol. The melting temperature at half-maximal value, $T_m$, was calculated using Protein Thermal Shift Software v1.4 (Applied Biosystems). Experiments were repeated in duplicate with similar results.

### Protein crystallization

Two collagen XVIII trimerization domains with introduced, inter-chain disulfide bond mutations, E31C–V'37C or G(-1)C–L'5C, were crystallized by sitting drop vapor diffusion. E31C–V'37C and G(-1)C–L'5C mutants at ~60 and ~52 mg·mL$^{-1}$, respectively, were mixed with reservoir solution at a 1:1 ratio to make 0.2 μL mother liquor drops using the MCSG-1 or MCSG-2 crystal screens (Anatrace). E31C–V'37C mutant crystals were grown in 100 mM Tris-HCl, pH 8.5, 200 mM lithium sulfate, 25% w/v PEG 3350 and formed overnight. Crystals were harvested after 8 days and required no further cryoprotection before being flash frozen in liquid nitrogen. G(-1) C–L'5C mutant crystals were grown in 100 mM Bis-Tris-HCl, pH 6.5, 25% w/v PEG 3350 and formed within 2 days. Crystals were harvested after ~14 days and required no further cryoprotection before being flash frozen in liquid nitrogen.

### X-Ray diffraction data collection and structure determination

X-ray diffraction data were collected at National Synchrotron Light Source II (NSLS-II) at Brookhaven National Laboratory (BNL) on beamline FMX 17-ID-2 for the E31C–V'37C collagen XVIII trimerization domain mutant, and on beamline NYX 19-ID for the G(-1)C–L'5C mutant. Diffraction data were processed with autoPROC[90] or XDS[91], respectively, and phasing was performed using Phenix[87] using Phaser-MR. The model was subjected to iterative rounds of refinement in phenix.refine and model building using Coot[89].

### ACE2 enzymatic activity assay

Following the instructions provided in the ACE2 enzymatic activity assay kit (Sigma-Aldrich catalog #MAK377-1KT), cleavage of a (7-methoxycoumarin-4-yl) acetyl-Ala-Pro-Lys(2,4-dinitrophenyl)-OH (Mca-APK-DNP) substrate was monitored to determine ACE2 enzymatic activity. T3A, T$^D$3A$^D$, T$^D$3A$^D_{R273Q}$, and T$^D$3A$^D_{H345F}$ were diluted to 1.25 μg·mL$^{-1}$ using the assay buffer. 50 μL of diluted protein, as well as positive and negative controls, were added to a 96-well plate and fluorescence was measured over 30 min after addition of substrate (320 nm excitation / 405 nm emission). BioTek Synergy H1 (Agilent) multimode plate reader was used to read plates. Experiments were repeated in duplicate with similar results.

### Computational methods

We performed all-atom MD simulations using NAMD[92] version 3.0-alpha-9 in Expanse supercomputer at the San Diego Supercomputing Center. The crystal structure of RBD-bound ACE2 (PDB ID: 6M0J) was used as a starting structure of 'wildtype' ACE2 for simulation. PyMOL[88] was used to generate the starting structure of the deactivation/disulfide mutants for

simulations. CHARMM-GUI[93] was used to generate simulation models, including the zinc ion. The CHARMM36 forcefield was applied to the system; we chose the TIP3 model for water. We simulated neutralizing the systems using KCl by adding an extra concentration of 50 mM to all simulations. We treated the long-range electrostatic interactions using PME and vDW using LJ potentials with a cutoff of 10 Å. After minimization, the systems were equilibrated at constant temperature (303 K) for 10 ns. Production simulations were then carried out for 400 ns using NPT ensemble simulations with timesteps of two femtoseconds. VMD tcl scripts[94] and the ProDy tool[95] were used to post-process and to calculate Root Mean Square Fluctuation and Root Mean Square Deviation. MD simulations for each construct were performed with $n = 1$.

### Serum stability assay

A Myc tag was introduced to the C-terminus of the T3A constructs, and an ELISA assay using anti-Myc antibody-horseradish peroxidase conjugate (anti-Myc-HRP, Thermo catalog #R951-25) was used to detect the antagonist. Stock tubes of purified T3A-Myc, T$^D$3A$^D$-Myc, T$^D$3A$^D_{R273Q}$-Myc and T$^D$3A$^D_{H345F}$-Myc (70 μg in 150 μL) were incubated in normal rat serum (Thermo) at 37 °C for up to 216 h. Aliquots (5 μL) from the stock tubes were removed at 0, 2, 4, 10, 24, 48, 72, 96, 120, 144, 168, 192, and 216 h and stored at −80 °C. The stability of the samples was determined via ELISA. A 96-well plate was coated with spike HexaPro protein (0.2 μg/well) overnight at 4 °C. The plate was blocked with 2% BSA in PBS for 1 h at room temperature. Samples were diluted in PBS (to ~1.2 nM) and incubated on a spike-coated plate in triplicate for 1 h at room temperature. The plate was washed 6 times with 1% TBST before anti-Myc-HRP antibody (1:5000) incubation on the plate for 1 h at room temperature. The plate was washed 6 times with 1% TBST, and SuperSignal Pico Chemiluminescence (Thermo) was added to the plates and read after 1 min shaking using BioTek Synergy H1 (Agilent) multimode plate reader. Plates were normalized to $t = 0$ min luminescence signal for 100% and a control sample luminescence signal without anti-Myc-HRP addition for 0%.

### Statistics and reproducibility

All pseudovirus, DSF, activity assays, and serum stability assays were run in triplicate with individual values graphed in figures. Triplicates are defined as three equal volumes aliquoted from a single stock and either measured directly (pseudovirus and DSF) or with aliquots taken over time (activity assay and serum stability). All error values are reported as standard deviation of $n$ biological replicates, as reported in each figure caption. Biological replicates are defined as separate experiments performed on separate days. Error values for SPR experiments are reported as the average standard deviation of $n = 2$ separate SPR experiments.

### Reporting summary

Further information on research design is available in the Nature Portfolio Reporting Summary linked to this article.

### Data availability

Crystallographic data for the structures reported in this article have been deposited in the Protein Data Bank (PDB) under deposition numbers: 9BNB and 9BNC. Single-particle electron microscopy data for the structures reported in the article have been deposited to the PDB and Electron Microscopy Data Bank (EMDB) (written as PDB/EMDB) under deposition numbers: 9BND/EMD-44724, 9BNE/EMD-44725, 9BNF/EMD-44726, and 9BNG/EMD-44727. Copies of this data can be obtained free of charge through https://www.rcsb.org/ and https://www.ebi.ac.uk/emdb/. All other relevant data generated and analyzed during the study are included in this article and supplementary information. Source data are provided with this paper (Supplementary Data 1–6).

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

## Acknowledgements

We are grateful to Songye Chen and the Caltech Cryo-EM facility for Cryo-EM data collection and Victor Garcia for Cryo-EM technical support. Cryo-EM was performed in the Beckman Institute Resource Center for Transmission Electron Microscopy at Caltech. We would also like to thank Priyanthi Gnanapragasam for the technical support of the pseudovirus assay and Dr. Christopher Barnes for select reagents. A component of this research used the AMX/FMX beamline of the National Synchrotron Light Source II, a U.S. Department of Energy (DOE) Office of Science User Facility operated for the DOE Office of Science by Brookhaven National Laboratory under Contract No. DE-SC0012704. The Center for BioMolecular Structure (CBMS) is primarily supported by the National Institutes of Health, National Institute of General Medical Sciences (NIGMS) through a Center Core P30 Grant (P30GM133893), and by the DOE Office of Biological and Environmental Research (KP1607011). Additional efforts were supported by Wellcome Leap (P.J.B.), the National Institutes of Health (P01-AI165075 (P.J.B.), Bill and Melinda Gates Foundation INV-034638 (P.J.B.), the Coalition for Epidemic Preparedness Innovations (CEPI) (P.J.B.), the Merkin Institute for Translational Research (Caltech). In addition, this work was supported by Institutional Funds (J.C.W.), Antibody Discovery Engine supported by the Integrated Drug Development Venture of City of Hope (J.C.W.) and the Drug Discovery and Structural Biology Shared Resource Facility at the City of Hope Comprehensive Cancer Center, supported by the National Cancer Institute of the National Institutes of Health under award number P30CA33572 (CCSG—John Carpten/J.C.W.). The content is solely the responsibility of the authors and does not necessarily represent the official views of the National Institutes of Health.

## Author contributions

J.G. and T.Y. designed and conducted biochemical, structural and in vitro experiments, analyzed the data, designed the figures and wrote the manuscript; H.C., Y.J., M.P., C.F., and R.P. conducted experiments and analyzed data; P.J.B. contributed to experiment planning and the manuscript. J.C.W. conceptualized and designed the overall experiments, analyzed data, wrote the manuscript, and secured funding. All authors approved the final version of the manuscript.

## Competing interests

The authors declare no competing interests.
