## [Transparent Peer review file · Communications Biology]

Development of an ultrahigh affinity, trimeric ACE2 biologic as a universal SARS-CoV-2 antagonist

Corresponding Author: Professor John Williams

This manuscript has been previously submitted at another journal. This document only contains information relating to versions considered at Communications Biology.

Version 0:

Reviewer comments:

Reviewer #1

(Remarks to the Author)

In this manuscript, the authors fuse the N-terminus of the monomeric protease domain of ACE2 to a small trimerization domain from collagen XVIII, producing a trimeric decoy receptor for highly avid binding to SARS-CoV-2 Spike trimers. The paper is well written with high quality figures and spans conception to biochemical characterization to cryo-EM and crystallographic structural analyses. While many of the concepts have been previously described, the research is well done and the novelty is sufficient for a journal like Communications Biology. The manuscript is lacking in a proper comparison to dimeric ACE2, and ideally would include some in vivo assessment, although I recognize that animal studies are easier said than done.

MAJOR COMMENTS

1. Comparisons are made to monomeric ACE2 as the control. This is the correct control for demonstrating the principles of trimerization and enhanced avidity, but next to nobody is developing monomeric ACE2 as a decoy receptor. Except in cases where ACE2 symmetry doesn't align with a fusion partner, in nearly all cases researchers are developing decoy receptors built upon dimeric ACE2 that contains the collectrin-like domain. This domain has multiple benefits:

- the collectrin-like domain enhances monovalent affinity of the protease domain for Spike (doi.org/10.1126/science.abc0870 and doi.org/10.1073/pnas.2016093117)
- the collectrin-like domain enhances pharmacokinetics (<https://www.science.org/doi/10.1126/sciadv.abq6527>)
- the ACE2 dimer will have avid inter-Spike binding on a virion surface

While the authors do discuss inter- versus intra-Spike avidity, it is quite possible that Spike density on SARS-CoV-2 is geometrically matched for avid binding to ACE2 dimers expressed on the surface of host cells for optimum infectivity. Inter-Spike binding to soluble ACE2 dimers may also be more effective in neutralization than intra-Spike binding to an ACE2 trimer, a point the authors do elaborate on in the Discussion. The authors have the tools to test this question by doing a head-to-head comparison of their ACE2 trimers versus an ACE2 dimer (normalized by number of binding sites), ideally with authentic virus that has the proper density of Spikes.

Likewise, the authors efforts to engineer stability might be moot in the case of ACE2 dimers, which are reported to have a T_m value closer to 60 deg C (doi.org/10.1371/journal.pone.0278294).

OTHER COMMENTS

2. In vitro serum stability may not correspond to in vivo serum stability. Ideally the authors would do a PK study, although I recognize this may not be possible at this stage of their research program.

3. The ACE2 trimers will almost certainly have poor PK. While their large size will reduce renal clearance, without a mechanism for recycling the protein will likely disappear from circulation quickly, irrespective of the thermal stability. This is

worth a discussion point, perhaps in the second to last paragraph where half-lives of various constructs are reported. Have the authors considered adding features for interacting with albumin?

4. It is worth noting that IgM fusions of ACE2 that are highly avid have also made it to human trials.

5. The authors describe "ACE2 coordinates the ACE2 substrate", but "coordinates" refers to a very specific kind of chemical bonding. I think their use of "coordinates"/"coordinating" in a couple instances is incorrect. ACE2 does coordinate a Zn²⁺ ion for metalloprotease activity with histidines 374 and 378, which are nearby H345. How is the Zn²⁺ handled in the MD simulations? Does Zn²⁺ remain at the active site? Mutations to the Zn²⁺ site have been shown to reduce T_m (doi.org/10.1371/journal.pone.0278294) and cause a decrease in serum half-life (doi.org/10.1016/j.omtm.2024.101301).

6. Figure captions at the end of the main text are different from figure captions below the images (at least for Figure 1). Double check which figure caption is the one intended.

7. In Table 1, I recommend reporting the KD in the last column in units of 10⁻⁹ (i.e. nanoM) or 10⁻¹² (i.e. picoM), rather than as 10⁻¹⁰ which is non-standard. It makes it easier to compare to values reported by others, which are often in nanoM.

Reviewer #2

(Remarks to the Author)

Gonzales and colleagues report the development of a high-affinity, soluble, enzymatically inactive, trimeric ACE2 as an inhibitor of SARS-CoV-2 entry. They describe the approach to the design and optimization of this inhibitor in a thorough and logical manner, including the evaluation of several scaffolds and linker compositions. They ultimately demonstrate a construct with high affinity binding to prefusion-stabilized S protein of SARS-CoV-2 and SARS-CoV(-1). This construct acts as an efficient inhibitor of ACE2-mediated viral entry in a pseudotyped lentivirus model. While the premise of a decoy receptor is not novel in itself, the particular technical approach to integrate optimization of affinity/avidity, stability, and enzymatic inactivity yields a novel and promising therapeutic candidate that will be of interest to the field.

The work is overall convincing, and methods are described in sufficient detail to facilitate reproducibility. Some of the data, however, are derived from an unclear number of independent observations versus technical replicates. For example, Fig. 1e,1f – is this one representative experiment or aggregate data from two independent experiments? The pertinent methods paragraph states, "Experiments were repeated in duplicate with similar results" but the figure legend is unclear on what is presented. When possible, figures should represent the aggregate of multiple experiments.

The serum stability assay is central to the investigators' proposal of clinical utility of this construct, but the results are challenging to interpret with insufficient timepoints to interpolate half-life. The experiment should ideally be repeated with additional measurements in the range of 100-200 hours. Also, please comment on how the Myc tag may influence stability.

It is not clear to this reviewer what the molecular dynamics simulation demonstrates or how it adds to the conclusions of the paper. Perhaps this could be better explained for readers less familiar with these computational techniques.

Minor comments:

"The Omicron variant, for instance, encodes over 50 known mutations¹² in the sequence of the spike protein compared to the original Wuhan-Hu-1 strain¹³⁻¹⁵." Spike protein or gene? Please clarify amino acid vs nucleotide substitutions.

"As the immune system weakens with age, the risk of mortality from COVID-19 dramatically increases^{16,17}, putting the elderly at greater risk." Advanced age is indeed a risk factor for mortality and complications of COVID-19, and immune function is not the only cause. The statement that "the immune system weakens with age" is a generalization that does not well support the authors' assertions.

Regarding long COVID, "no current treatment available" – there is growing evidence basis for symptomatic and supportive treatments as reviewed in "Long COVID: major findings, mechanisms and recommendations" (Davis et al.).

Paragraph 4 ("Whereas vaccines are prophylactic...") – this section is unclear and should be rewritten; "reduce the patient's immune system"? Note that there have been non-vaccine prophylaxis options for COVID-19 as well, e.g. Evusheld.

"differential scanning fluorescence" – should read "differential scanning fluorimetry"

Final discussion paragraph, "this decoy antagonist will be immediately useful for immunocompromised patients" – understates the extensive further study and optimization needed for commercialization and clinical application.

Methods section, DSF – final concentration of SYPRO Orange is unclear.

Reviewer #3

(Remarks to the Author)

Comments:

In this manuscript, Juliet Gonzales et al. sought a trimeric scaffold collagen XVIII to create an ultrapotent, trivalent ACE2 entry antagonist. Distinct disulfide bonds E31C–V'37C and N51C–V343C were added in the collagen XVIII trimerization domain and ACE2, respectively, to increase the overall thermal and serum stability. A single point mutation of R273Q was introduced to inhibit ACE2 enzyme activity. The enzymatically inactivated construct, TD3ADR273Q, closely matches the C3 symmetry of the spike. It binds to the spike with high affinity, exhibits potent viral inhibition against all tested SARS-CoV-2 variants, and shows remarkable stability in serum. TD3ADR273Q was expected to become the candidate widely effective against SARS-CoV-2 variants and other ACE2-utilizing entry viruses.

Specific comments:

1. The authors thought that the trimeric ACE2 decoy antagonist could widely inhibit SARS-CoV-2 variants and other ACE2-utilizing entry viruses. The SPR results of this paper has demonstrated that the trimeric ACE2 antagonist, TD3ADR273Q, exhibits a strong binding affinity to SARS-CoV-2 and SARS-CoV. TD3ADR273Q can inhibit the pseudovirus of SARS-CoV-2 and its variants fusion. However, there is a lack of experimental data on TD3ADR273Q's ability to neutralize SARS-CoV pseudovirus. Could you please provide additional data for this part of the experiment?

2. Regarding the SPR results in Figure 1, please add the KD values to facilitate direct comparison of the binding affinities among different groups.

3. The researchers mention that the standard deviation ranges observed in the ACE2 monomer-bound structures are 0.8-10 Å. Is this concept widely accepted? Have you personally summarized this concept, or is it proposed in another research paper? If it is based on your own experience, please provide an explanation. If it is from a published paper, please cite the source.

Version 1:

Reviewer comments:

Reviewer #1

(Remarks to the Author)

While more experiments focused on in vivo PK and direct comparisons to prior ACE2 decoy receptors would have been ideal, the manuscript is nonetheless a full description of the design of a symmetry-matched decoy that covers conception, optimization, biochemical characterization, and structure determination. I am also sensitive to the difficulties in completing the desired studies in the current environment. I therefore recommend the manuscript's publication without further revision.

Reviewer #2

(Remarks to the Author)

The authors have addressed all concerns raised in the initial review. The additional experiments incorporated into Fig 3i clearly demonstrate the differences in serum stability among the generated constructs. The expanded explanation of the premise and findings of the molecular dynamics simulation will be appreciated by readers with a non-structural biology background. I congratulate the authors on their strong work.

Reviewer #3

(Remarks to the Author)

The authors have addressed my concerns and incorporated my suggestions satisfactorily. I approve this manuscript for publication.

RE: Reviewer's comments

We are extremely grateful to the reviewers for their time, their overall positive comments on our work, and their insightful critiques. We have addressed these to the best of our ability, given that the primary student working on this project has moved on, other lab members have overlapping commitments to their own projects, and financial constraints.

We emphasize that the work presented here reflects a design concept—recognizing the symmetry of the spike trimer, identifying an ultra-stable human trimeric scaffold, and optimizing the linker length. Moreover, we provide definitive proof of the design using cryo-EM as well as improving a trimeric, ultra stable scaffold – also confirmed by a high resolution X-ray structure.

We fully agree with the reviewers that additional work is needed to advance this “development candidate” to a clinical candidate, including extensive animal studies. Unfortunately, we currently lack the funding for this next phase. We believe that publishing these results will enable us to continue this work and, ideally, attract the attention of a biotech or pharmaceutical company that can provide the necessary resources to develop this into a viable therapeutic.

Notably, unlike antibodies and other engineering efforts that focus solely on binding the current CoV-2 RBD, our symmetry-based ACE2 biologic potently blocks multiple strains of CoV-2 as well as CoV-1. By design, we fully anticipate it will also block future CoV-2 variants or other emerging ACE2-based coronaviruses.

Whether other ACE2 designs suggested by the reviewers ultimately outperform ours (after CMC efforts) can only be determined through clinical studies. We sincerely hope we have addressed the reviewers' concerns and that this work can now be made available to the scientific community.

Sincerely,

John Williams

Reviewer #1 (Remarks to the Author): **Our responses are in green.**

In this manuscript, the authors fuse the N-terminus of the monomeric protease domain of ACE2 to a small trimerization domain from collagen XVIII, producing a trimeric decoy receptor for highly avid binding to SARS-CoV-2 Spike trimers. The paper is well written with high quality figures and spans conception to biochemical characterization to cryo-EM and crystallographic structural analyses. While many of the concepts have been previously described, the research is well done and the novelty is sufficient for a journal like Communications Biology. The manuscript is lacking in a proper comparison to dimeric ACE2, and ideally would include some in vivo assessment, although I recognize that animal studies are easier said than done.

MAJOR COMMENTS

1. Comparisons are made to monomeric ACE2 as the control. This is the correct control for demonstrating the principles of trimerization and enhanced avidity, but next to nobody is developing monomeric ACE2 as a decoy receptor. Except in cases where ACE2 symmetry doesn't align with a fusion partner, in nearly all cases researchers are developing decoy receptors built upon dimeric ACE2 that contains the collectrin-like domain. This domain has multiple benefits:

- the collectrin-like domain enhances monovalent affinity of the protease domain for Spike (doi.org/10.1126/science.abc0870 and doi.org/10.1073/pnas.2016093117)
- the collectrin-like domain enhances pharmacokinetics (<https://www.science.org/doi/10.1126/sciadv.abq6527>)
- the ACE2 dimer will have avid inter-Spike binding on a virion surface

While the authors do discuss inter- versus intra-Spike avidity, it is quite possible that Spike density on SARS-CoV-2 is geometrically matched for avid binding to ACE2 dimers expressed on the surface of host cells for optimum infectivity. Inter-Spike binding to soluble ACE2 dimers may also be more effective in neutralization than intra-Spike binding to an ACE2 trimer, a point the authors do elaborate on in the Discussion. The authors have the tools to test this question by doing a head-to-head comparison of their ACE2 trimers versus an ACE2 dimer (normalized by number of binding sites), ideally with authentic virus that has the proper density of Spikes.

Likewise, the authors efforts to engineer stability might be moot in the case of ACE2 dimers, which are reported to have a T_m value closer to 60 deg C (doi.org/10.1371/journal.pone.0278294).

We appreciate the reviewer's comments and agree that comparison to dimeric inhibitors is useful. However, we suggest that such a comparison would only be meaningful in an extensive study using animal models. Notably, the primary focus of this manuscript is the recognition of the C3 symmetry in the spike protein and how this can be used to optimize affinity. We also

recognize that there are other means to improve overall affinity, including adding the collectrin-like domain, other fusions, bivalent interactions, etc. We are not claiming that our approach is the best approach.

Given this, we have respectfully not performed these direct comparisons. First, since the conception of this approach, the ability to conduct meaningful animal studies has been effectively impossible. Capable PIs and labs with whom we could collaborate were booked out for years in the early days of this project. The costs were prohibitive for anything short of a biotech company or an expert COVID lab. Additionally, the emergence of new variants constantly moved the goalposts. Furthermore, the influx of labs entering the space (including ours) significantly impacted our ability to secure funding to go beyond design principles.

On the other hand, we were able to run pseudoviral assays covering multiple variants (with much gratitude to the Bjorkman group). While these pseudoviral assays were very effective in characterizing our constructs, they are limited and do not represent actual SARS-CoV-1/2 viruses. Thus, fine distinctions between different ACE2 fusion formats (e.g., pentamer, etc.) using this assay are unlikely to yield meaningful differences. Again, extensive animal studies with multiple variants are truly needed. We are seeking funding to pursue this, but it remains a challenge.

Regarding the reviewer's comment that spike density may be optimized to bind dimeric ACE2, we tend to agree—it could be. However, it may not be fully optimized. The presence of multiple ACE2 receptors on a cell surface and multiple ligands (RBDs) effectively leads to avidity. In this scenario, multiple, distant (nonadjacent) monomeric interactions are likely sufficient for engagement and internalization. Neither scenario detracts from the design elements presented herein—using the inherent symmetry of the spike protein ensures an extremely high local concentration of ACE2 at the RBDs, ensuring each will be occupied. This also allows for accommodation of point mutations that reduce the strength of the monomeric ACE2-RBD interaction. We have added to the manuscript to address this point.

Regarding the stability of ACE2 (“authors’ efforts to engineer stability might be moot in the case of ACE2 dimers”), we note that, based on our experience translating biologics into the clinic (JCW is a co-founder of Xilio Therapeutics), significant efforts during CMC are invariably spent on improving multiple properties, including thermal stability, of every lead development candidate to produce a clinical candidate. Stability encompasses multiple properties, including protease resistance, and is always part of a Target Product Profile. Moreover, ACE2 (dimeric) is continuously expressed by the cell and is membrane-bound. The selection pressure is different for endogenous and active ACE2 compared to an ACE2-based therapeutic. Finally, our data clearly indicate that the introduction of an intramolecular disulfide improved thermal stability.

OTHER COMMENTS

2. In vitro serum stability may not correspond to in vivo serum stability. Ideally the authors would do a PK study, although I recognize this may not be possible at this stage of their research program.

We completely agree. In vitro serum stability is a low bar, but it is also an essential one to pass. We also acknowledge that proper PK studies in animals are challenging. Ideally, these would be conducted in both non-infected and infected animals. We agree with the reviewer's comments (vide infra) that we will likely need to add a lifetime extension moiety (e.g., albumin, FcRn interactors). These are very expensive experiments and require additional sources of funding, which we are actively pursuing. Again, the primary focus of this manuscript is the design principle.

3. The ACE2 trimers will almost certainly have poor PK. While their large size will reduce renal clearance, without a mechanism for recycling the protein will likely disappear from circulation quickly, irrespective of the thermal stability. This is worth a discussion point, perhaps in the second to last paragraph where half-lives of various constructs are reported. Have the authors considered adding features for interacting with albumin?

As noted above, we completely agree. We have added comments vis-à-vis this concern in the discussion.

4. It is worth noting that IgM fusions of ACE2 that are highly avid have also made it to human trials.

Thank you for the suggestion. We have added a line in the introduction describing and referencing this study. That said, we maintain that our approach is intraspikes, whereas the IgM approach is likely interspike. Again, we are not arguing that our approach is superior—it is fundamentally different. Also relevant to some of the previous comments, IgMs are rapidly cleared as they do not interact with the FcRn receptor. However, there was sufficient evidence to move the IgM approach into human trials despite its short half-life.

5. The authors describe "ACE2 coordinates the ACE2 substrate", but "coordinates" refers to a very specific kind of chemical bonding. I think their use of "coordinates"/"coordinating" in a couple instances is incorrect. ACE2 does coordinate a Zn²⁺ ion for metalloprotease activity with histidines 374 and 378, which are nearby H345. How is the Zn²⁺ handled in the MD simulations? Does Zn²⁺ remain at the active site? Mutations to the Zn²⁺ site have been shown to reduce T_m (doi.org/10.1371/journal.pone.0278294) and cause a decrease in serum half-life (doi.org/10.1016/j.omtm.2024.101301).

We apologize for the confusion. First, we now describe the parameters of the simulation in more detail, including the energy parameters for Zn²⁺, in the Methods section. Additionally, we used the term "coordinate" more colloquially (e.g., positioning a substrate) than we should have, without intending the more precise usage from inorganic chemistry. We have removed this

potentially confusing language and now describe the role of H345 in substrate binding more clearly in the Results section.

6. Figure captions at the end of the main text are different from figure captions below the images (at least for Figure 1). Double check which figure caption is the one intended.

We deeply apologize for the mistake. We have addressed this.

7. In Table 1, I recommend reporting the KD in the last column in units of 10^{-9} (i.e. nanoM) or 10^{-12} (i.e. picoM), rather than as 10^{-10} which is non-standard. It makes it easier to compare to values reported by others, which are often in nanoM.

We agree and have addressed this.

Reviewer #2 (Remarks to the Author):

Gonzales and colleagues report the development of a high-affinity, soluble, enzymatically inactive, trimeric ACE2 as an inhibitor of SARS-CoV-2 entry. They describe the approach to the design and optimization of this inhibitor in a thorough and logical manner, including the evaluation of several scaffolds and linker compositions. They ultimately demonstrate a construct with high affinity binding to prefusion-stabilized S protein of SARS-CoV-2 and SARS-CoV(-1). This construct acts as an efficient inhibitor of ACE2-mediated viral entry in a pseudotyped lentivirus model. While the premise of a decoy receptor is not novel in itself, the particular technical approach to integrate optimization of affinity/avidity, stability, and enzymatic inactivity yields a novel and promising therapeutic candidate that will be of interest to the field.

We appreciate the reviewer's positive response.

The work is overall convincing, and methods are described in sufficient detail to facilitate reproducibility. Some of the data, however, are derived from an unclear number of independent observations versus technical replicates. For example, Fig. 1e,1f – is this one representative experiment or aggregate data from two independent experiments? The pertinent methods paragraph states, “Experiments were repeated in duplicate with similar results” but the figure legend is unclear on what is presented. When possible, figures should represent the aggregate of multiple experiments.

We apologize the omission. We have addressed this in the manuscript.

The serum stability assay is central to the investigators' proposal of clinical utility of this construct, but the results are challenging to interpret with insufficient timepoints to interpolate half-life. The experiment should ideally be repeated with additional measurements in the range of 100-200 hours. Also, please comment on how the Myc tag may influence stability.

We agree. We have rerun the serum stability experiment and addressed this in the manuscript. Regarding the Myc-tag, there is still significant work required to advance this “design principle” into a clinical candidate, as noted above. This includes removal of the Myc-tag (which served as

a tractable handle for preclinical development), potential addition of a half-life extension tag, and other modifications necessary for clinical development.

It is not clear to this reviewer what the molecular dynamics simulation demonstrates or how it adds to the conclusions of the paper. Perhaps this could be better explained for readers less familiar with these computational techniques.

We understand the reviewer's point and acknowledge that we failed to be more explicit about the rationale in the original manuscript. Briefly, the MD simulations were performed to better understand the effects of the mutations, as we observed differences in the serum and thermal stability experiments. We also aimed to identify additional positions that could potentially be exploited to enhance overall stability.

Given our experience using MD to support the humanization of monoclonal antibodies and to design "bionics" (<https://doi.org/10.1182/blood.2023021570>), these were relatively straightforward calculations to perform. As noted in the manuscript, we observed a correlation between serum stability (DSF) and RMSD calculations. We also found that the "parental" ACE2 was the most dynamic. While we did not identify obvious positions for further modification, the simulations provided useful insights and can help guide the development of a clinical candidate.

Regardless, we thank the reviewer for the comment and have revised the manuscript to better justify and clarify this section.

Minor comments:

"The Omicron variant, for instance, encodes over 50 known mutations¹² in the sequence of the spike protein compared to the original Wuhan-Hu-1 strain¹³⁻¹⁵." Spike protein or gene? Please clarify amino acid vs nucleotide substitutions.

Thank you! We have added this to the manuscript: "The Omicron variant, for instance, encodes over 50 known amino acid mutations¹² in the spike protein compared to the original Wuhan-Hu-1 strain¹³⁻¹⁵."

"As the immune system weakens with age, the risk of mortality from COVID-19 dramatically increases^{16,17}, putting the elderly at greater risk." Advanced age is indeed a risk factor for mortality and complications of COVID-19, and immune function is not the only cause. The statement that "the immune system weakens with age" is a generalization that does not well support the authors' assertions.

We absolutely agree and apologize for the overly simplistic, incomplete statement. We have addressed this in the revised manuscript: "Mortality from COVID-19 dramatically increases with age^{16,17} and for those with compromised immune systems^{18,19}. A study comparing COVID-19-related hospitalizations, conducted from January 2022 to December 2022, found that immunocompromised individuals comprised 3.9% of the study population, but accounted for 24% of COVID-19-related deaths²⁰."

Regarding long COVID, “no current treatment available” – there is growing evidence basis for symptomatic and supportive treatments as reviewed in “Long COVID: major findings, mechanisms and recommendations” (Davis et al.).

We are grateful for the reviewer’s insight and comments. We have addressed this in the manuscript.

Paragraph 4 (“Whereas vaccines are prophylactic...”) – this section is unclear and should be rewritten; “reduce the patient’s immune system”? Note that there have been non-vaccine prophylaxis options for COVID-19 as well, e.g. Evusheld.

We understand the reviewer’s concern (“reduce the patient’s immune system?”) and have attempted to provide more clarity. The intention was to convey that anti-inflammatories are administered to patients to reduce the impact of cytokine release syndrome (CRS), which results from a hyperactive or dysregulated immune response.

We also acknowledge the reviewer’s point that Evusheld was used as a prophylactic. It has since been replaced by Pemgarda, a monoclonal antibody that has maintained activity against the latest XEC variant. We have incorporated this suggestion into the manuscript.

The overall intent of this paragraph is to motivate the development of a pan-CoV inhibitor

“differential scanning fluorescence” – should read “differential scanning fluorimetry”

Thank you for bringing this to our attention. It’s been corrected in the manuscript.

Final discussion paragraph, “this decoy antagonist will be immediately useful for immunocompromised patients” – understates the extensive further study and optimization needed for commercialization and clinical application.

We completely agree. As noted above, this manuscript aims to confirm an observation (e.g., symmetry) and implement a design principle (geometry and avidity). We argue that we have achieved this. The development of a novel, bona fide clinical candidate—complete with animal experiments, appropriate controls, and head-to-head comparisons with other modalities (e.g., ACE2-Fc)—is a major undertaking and would constitute a separate manuscript. Regardless, we have modified the text to address this comment.

Methods section, DSF – final concentration of SYPRO Orange is unclear.

We have added experimental details to the manuscript.

Reviewer #3 (Remarks to the Author):

Comments:

In this manuscript, Juliet Gonzales et al. sought a trimeric scaffold collagen XVIII to create an ultrapotent, trivalent ACE2 entry antagonist. Distinct disulfide bonds E31C–V'37C and N51C–V343C were added in the collagen XVIII trimerization domain and ACE2, respectively, to increase the overall thermal and serum stability. A single point mutation of R273Q was introduced to inhibit ACE2 enzyme activity. The enzymatically inactivated construct, TD3ADR273Q, closely matches the C3 symmetry of the spike. It binds to the spike with high affinity, exhibits potent viral inhibition against all tested SARS-CoV-2 variants, and shows remarkable stability in serum. TD3ADR273Q was expected to become the candidate widely effective against SARS-CoV-2 variants and other ACE2-utilizing entry viruses.

Specific comments:

1. The authors thought that the trimeric ACE2 decoy antagonist could widely inhibit SARS-CoV-2 variants and other ACE2-utilizing entry viruses. The SPR results of this paper has demonstrated that the trimeric ACE2 antagonist, TD3ADR273Q, exhibits a strong binding affinity to SARS-CoV-2 and SARS-CoV. TD3ADR273Q can inhibit the pseudovirus of SARS-CoV-2 and its variants fusion. However, there is a lack of experimental data on TD3ADR273Q's ability to neutralize SARS-CoV pseudovirus. Could you please provide additional data for this part of the experiment?

We absolutely agree and have done so. We are pleased to report that the trimeric scaffold also inhibits SARS-CoV-1 in pseudoviral assays. We are grateful for the reviewer's suggestion, as it supports the idea that this approach is highly likely to be applicable as a viral inhibitor against future, unknown SARS-CoV-x strains.

2. Regarding the SPR results in Figure 1, please add the KD values to facilitate direct comparison of the binding affinities among different groups.

Thank you for the suggestion. Data was originally reported in Table 1 only, but we added these values directly to Figure 1 for clarity.

3. The researchers mention that the standard deviation ranges observed in the ACE2 monomer-bound structures are 0.8-10 Å. Is this concept widely accepted? Have you personally summarized this concept, or is it proposed in another research paper? If it is based on your own experience, please provide an explanation. If it is from a published paper, please cite the source.

Thank you for asking for clarification. We have re-written this paragraph for clarity.

"We then attempted to evaluate the 'symmetry of interaction' between our ACE2 constructs with spike protein (T0A-S, T3A-S, T5A-S, and T18A-S; where S is the SARS-CoV-2 spike Hexaprotein).

protein) by likening the C3 symmetry to an equilateral triangle with all sides being equal length and a corresponding standard deviation between these lengths of zero (s.d. = 0). Thus, we calculated lengths (distances) and corresponding standard deviations between the C α atoms of equivalent residues in the RBDs of the spike protein (residue D428; as determined by Fan, et al. 2022⁷⁰) and in each of the ACE2 domains (residue N137; arbitrarily determined residue on ACE2) (Fig. 2, right, Table 3). The calculated mean distances and s.d. between spike RBD were: 41 \pm 3.6 Å for T0A-S; 40 \pm 1.0 Å for T3A-S; 40 \pm 1.0 Å for T5A-S; and 42 \pm 3.2 Å for T18A-S. Equating lower s.d. values to symmetry, we determine T3A and T5A to have higher symmetry than T0A or T18A. Next, we calculated mean distance and s.d. between ACE2: 149 \pm 13.2 Å for T0A-S; 141 \pm 7.6 Å for T3A-S; 144 \pm 11.2 Å for T5A-S; and 142 \pm 10.8 Å for T18A-S. Again, equating lower s.d. values to symmetry, we now determine T3A to have the highest symmetry of binding between constructs. These distances and s.d. were compared to those of five cryo-EM structures of spike trimer with three ACE2 monomers bound (see Methods) to compare symmetry and determine whether binding of our constructs potentially force the RBD domains into un-natural or energetically unfavorable conformations. We observed average mean inter-RBD distances of 41 \pm 1.9 Å and mean inter-ACE2 distances of 144 \pm 6.0 Å. As expected, the s.d. values of inter-RBD and inter-ACE2 of these models without a trimeric domain are similar or lower than any of our complexes – indicating higher order symmetry. Regardless, the mean distances between spike RBD or ACE2 in each of our constructs fall within one standard deviation of that calculated from published structures and suggests that TXA constructs do not require or induce energetically unfavorable spike RBD conformations for interactions. Combined with clear density for the collagen XVIII trimerization domain and ‘optimal’ positioning of individual ACE2 domains, the T3A represented the ‘best’ compromise and was selected for further development.”

Reviewers' comments:

We are extremely grateful to the reviewers for their time, overall positive comments on work, and their insightful criticism. We have addressed these to the best of ability given time and financial restraints. We emphasize that the work presented here reflects a design concept – recognizing the symmetry of the spike trimer, identification of ultra stable, human trimeric scaffold and the optimization of the linker length. Moreover, we provide definitive proof of the design using cryoEM. We fully agree with the reviewers that additional work needs to be done to turn this “development candidate” to a clinical candidate including extensive animal studies. We simply don't have funding for this and believe the publication of these results will allow us to continue or even better catch the attention of a biotech and/or pharma to bring all the resources needed to make this a viable therapeutic. Notably, unlike antibodies and other engineering efforts that uniquely focus on binding the current CoV2 RDB, we demonstrate our symmetry based, ACE2 biologic potentially blocks multiple strains of CoV2 as well as Cov1. By design, we fully anticipate it will block future CoV2 variants or emerging/future ACE2-based Covid strains. Whether other ACE2 designs suggested by the reviewers outperform our design (after CMC efforts) ultimately requires clinical studies. We sincerely hope we have addressed the reviewers concerns and are able to make this work available to scientific community.

Reviewer #1 (Remarks to the Author):

In this manuscript, the authors fuse the N-terminus of the monomeric protease domain of ACE2 to a small trimerization domain from collagen XVIII, producing a trimeric decoy receptor for highly avid binding to SARS-CoV-2 Spike trimers. The paper is well written with high quality figures and spans conception to biochemical characterization to cryo-EM and crystallographic structural analyses. While many of the concepts have been previously described, the research is well done and the novelty is sufficient for a journal like Communications Biology. The manuscript is lacking in a proper comparison to dimeric ACE2, and ideally would include some in vivo assessment, although I recognize that animal studies are easier said than done.

MAJOR COMMENTS

1. Comparisons are made to monomeric ACE2 as the control. This is the correct control for demonstrating the principles of trimerization and enhanced avidity, but next to nobody is developing monomeric ACE2 as a decoy receptor. Except in cases where ACE2 symmetry doesn't align with a fusion partner, in nearly all cases researchers are developing decoy receptors built upon dimeric ACE2 that contains the collectrin-like domain. This domain has multiple benefits:

- the collectrin-like domain enhances monovalent affinity of the protease domain for Spike (doi.org/10.1126/science.abc0870 and doi.org/10.1073/pnas.2016093117)
- the collectrin-like domain enhances pharmacokinetics (<https://www.science.org/doi/10.1126/sciadv.abq6527>)
- the ACE2 dimer will have avid inter-Spike binding on a virion surface

While the authors do discuss inter- versus intra-Spike avidity, it is quite possible that Spike density on SARS-CoV-2 is geometrically matched for avid binding to ACE2 dimers expressed on the surface of host cells for optimum infectivity. Inter-Spike binding to soluble ACE2 dimers may also be more effective in neutralization than intra-Spike binding to an ACE2 trimer, a point the authors do elaborate on in the Discussion. The authors have the tools to test this question by doing a head-to-head comparison of their ACE2 trimers versus an ACE2 dimer (normalized by number of binding sites), ideally with authentic virus that has the proper density of Spikes.

Likewise, the authors efforts to engineer stability might be moot in the case of ACE2 dimers, which are reported to have a T_m value closer to 60 deg C (doi.org/10.1371/journal.pone.0278294).

We appreciate the reviewer's comments and agree that comparison to dimeric inhibitors is useful. However, we suggest this comparison would only pan out in an extensive study using animal models. Notably, the primary focus of this manuscript is the recognition of the C3 symmetry in the spike protein and that this can be used to optimize the affinity. We also recognize that there are other means to improve the overall affinity including adding the collectrin-like domain, other fusions, bivalent interactions, etc. We are not making the claim that our approach is the best approach.

Given this, we respectfully have not preformed these direct comparisons. First, since the conception of this approach, the ability to do meaningful animal studies has been effectively impossible. Capable PIs and labs that we could collaborate with were booked out years. The costs were prohibitive for anything short of a biotech company. Additional variants moved the goal posts constantly. In addition, the swell of labs entering the space (including ours) significantly impacted adequate funding.

On the other hand, we were able to run pseudoviral assays covering multiple variants (with much gratitude to the Bjorkman group). While these pseudoviral assays were very good at characterizing our constructs, they are limited and certainly not the actual SARS-CoV-1/2 viruses. Thus, fine slicing differences between other formats of ACE2 fusions (e.g., pentamer, etc.) using this pseudoviral assay is very unlikely to establish meaningful differences. Again, extensive animal studies with multiple variants are really needed. We are seeking funding to do so, but this remains challenging.

Per the reviewer's comment that the spike density is potentially optimized to bind dimeric ACE2, we tend to agree – it could be. It may not be optimized. Having multiple ACE2 receptors on a cell surface and multiple ligands (RBDs) affectively leads to avidity. In this scenario, multiple, distant (nonadjacent) monomeric interactions are likely sufficient for engagement and internalization. Neither scenario detracts from the design elements presented herein – using the inherent symmetry of the spike protein ensures an extremely high local concentration of ACE2 to the RBDs, ensuring each will be occupied. This also ensures point mutations that effectively reduce the strength of the monomeric ACE2-RBD interaction can be accommodated. We have added to the manuscript to address this point.

Per stability of the ACE2 (“authors efforts to engineer stability might be moot in the case of ACE2 dimers”), being heavily involved in translating biologics into the clinic (JCW is a co-Founder of Xilio Therapeutics) significant efforts during CMC are invariably spent on thermal stability of every lead development candidate to produce a clinical candidate. The stability “captures” multiple properties including protease resistance and always part of a Target Product

Profile. Moreover, ACE2 (dimeric) is continuously expressed by the cell and is membrane bound. The selection pressure is different for endogenous and active ACE2 compared to a ACE2-based therapeutic. Moreover, our data clearly indicates that the introduction of a disulfide improved the thermal stability.

OTHER COMMENTS

2. In vitro serum stability may not correspond to in vivo serum stability. Ideally the authors would do a PK study, although I recognize this may not be possible at this stage of their research program.

We completely agree. In vitro serum stability is a low bar but also an essential bar to pass. We also submit that proper PK studies in animals is challenging. Ideally, these would be done with non-infected and with infected animals. We also agree with the reviewer's comments (vida infra) that we likely need to add a lifetime extension moiety (e.g., Albumin, FcRn interactors). These are very expensive experiments and require additional sources of funding which we are actively pursuing. Again, the primary focus of this manuscript is the design principle.

3. The ACE2 trimers will almost certainly have poor PK. While their large size will reduce renal clearance, without a mechanism for recycling the protein will likely disappear from circulation quickly, irrespective of the thermal stability. This is worth a discussion point, perhaps in the second to last paragraph where half-lives of various constructs are reported. Have the authors considered adding features for interacting with albumin?

As noted above, we completely agree. We have added comments vis-à-vis this concern in the discussion.

4. It is worth noting that IgM fusions of ACE2 that are highly avid have also made it to human trials.

Thank you for the suggestion. We added a line in the intro describing and referencing this study. That said, we maintain that our approach would be intraspine whereas the IgM approach is likely to be interspike. Again, we are not arguing our approach is superior, its fundamentally different. Also relevant to some of the previous comments, IgMs are rapidly cleared as they do not interact with the FcRn receptor. However, there was sufficient evidence to move this IgM approach to human trials despite the short half-life.

5. The authors describe "ACE2 coordinates the ACE2 substrate", but "coordinates" refers to a very specific kind of chemical bonding. I think their use of "coordinates"/"coordinating" in a couple instances is incorrect. ACE2 does coordinate a Zn²⁺ ion for metalloprotease activity with histidines 374 and 378, which are nearby H345. How is the Zn²⁺ handled in the MD simulations? Does Zn²⁺ remain at the active site? Mutations to the Zn²⁺ site have been shown to reduce T_m (doi.org/10.1371/journal.pone.0278294) and cause a decrease in serum half-life (doi.org/10.1016/j.omtm.2024.101301).

We apologize for the confusion. First, we describe in more detail the parameters of the simulation including the energy parameters for the Zn²⁺ in the methods. Also, we used "coordinate" more colloquially (e.g., positioning a substrate) than we should, not intending the more precise usage in inorganic chemistry. We have removed this confusing language and described the role of H345 in binding its substrate in the results section.

6. Figure captions at the end of the main text are different from figure captions below the images (at least for Figure 1). Double check which figure caption is the one intended.

We deeply apologize for the mistake. We have addressed this.

7. In Table 1, I recommend reporting the KD in the last column in units of 10^{-9} (i.e. nanoM) or 10^{-12} (i.e. picoM), rather than as 10^{-10} which is non-standard. It makes it easier to compare to values reported by others, which are often in nanoM.

We agree and have addressed this.

Reviewer #2 (Remarks to the Author):

Gonzales and colleagues report the development of a high-affinity, soluble, enzymatically inactive, trimeric ACE2 as an inhibitor of SARS-CoV-2 entry. They describe the approach to the design and optimization of this inhibitor in a thorough and logical manner, including the evaluation of several scaffolds and linker compositions. They ultimately demonstrate a construct with high affinity binding to prefusion-stabilized S protein of SARS-CoV-2 and SARS-CoV(-1). This construct acts as an efficient inhibitor of ACE2-mediated viral entry in a pseudotyped lentivirus model. While the premise of a decoy receptor is not novel in itself, the particular technical approach to integrate optimization of affinity/avidity, stability, and enzymatic inactivity yields a novel and promising therapeutic candidate that will be of interest to the field.

We appreciate the reviewer's positive response.

The work is overall convincing, and methods are described in sufficient detail to facilitate reproducibility. Some of the data, however, are derived from an unclear number of independent observations versus technical replicates. For example, Fig. 1e,1f – is this one representative experiment or aggregate data from two independent experiments? The pertinent methods paragraph states, “Experiments were repeated in duplicate with similar results” but the figure legend is unclear on what is presented. When possible, figures should represent the aggregate of multiple experiments.

We apologize the omission. We have addressed this in the manuscript.

The serum stability assay is central to the investigators' proposal of clinical utility of this construct, but the results are challenging to interpret with insufficient timepoints to interpolate half-life. The experiment should ideally be repeated with additional measurements in the range of 100-200 hours. Also, please comment on how the Myc tag may influence stability.

We agree. We have rerun the serum stability experiment and addressed this in the manuscript. Per the Myc-tag, there is much work left to turn this “design principle” to a clinical candidate as noted above, including the removal of the Myc-tag (a tractable handle for “preclinical” development of this approach), potential addition of lifetime tag, and other additional modifications needed for a clinical candidate.

It is not clear to this reviewer what the molecular dynamics simulation demonstrates or how it adds to the conclusions of the paper. Perhaps this could be better explained for readers less familiar with these computational techniques.

We understand the reviewer's point and failed to be more explicit on the "why" in the original manuscript. Briefly, the MD simulations were performed to better understand the effect of the mutations as we observed differences in the serum and thermal stability experiments. We also wanted to identify additional positions that could be potentially exploited to enhance the overall stability. Given our experience using MD to help drive the humanization of monoclonal antibodies as well as to design "bionics" (<https://doi.org/10.1182/blood.2023021570>), these were relatively straightforward calculations to be performed. As noted in the manuscript we did observe a correlation between the serum stability (DSF) and the RMSD calculations. We also observe the "parental" ACE2 was the most dynamic. While we did not observe "obvious" positions to modify, these simulations did give insight and can be used to in part to guide the development of a clinical candidate. Regardless, we thank the reviewer for the comment and have modified the manuscript with the goal of justifying and adding clarity to this section.

Minor comments:

"The Omicron variant, for instance, encodes over 50 known mutations¹² in the sequence of the spike protein compared to the original Wuhan-Hu-1 strain¹³⁻¹⁵." Spike protein or gene? Please clarify amino acid vs nucleotide substitutions.

Thank you! We have added this to the manuscript: "The Omicron variant, for instance, encodes over 50 known amino acid mutations¹² in the spike protein compared to the original Wuhan-Hu-1 strain¹³⁻¹⁵."

"As the immune system weakens with age, the risk of mortality from COVID-19 dramatically increases^{16,17}, putting the elderly at greater risk." Advanced age is indeed a risk factor for mortality and complications of COVID-19, and immune function is not the only cause. The statement that "the immune system weakens with age" is a generalization that does not well support the authors' assertions.

We absolutely agree and apologize for the overly simplistic, incomplete statement. We have addressed this in the revised manuscript: "Mortality from COVID-19 dramatically increases with age^{16,17} and for those with compromised immune systems^{18,19}. A study comparing COVID-19-related hospitalizations, conducted from January 2022 to December 2022, found that immunocompromised individuals comprised 3.9% of the study population, but accounted for 24% of COVID-19-related deaths²⁰."

Regarding long COVID, "no current treatment available" – there is growing evidence basis for symptomatic and supportive treatments as reviewed in "Long COVID: major findings, mechanisms and recommendations" (Davis et al.).

We are grateful for the reviewer's insight and comments. We have addressed this in the manuscript.

Paragraph 4 ("Whereas vaccines are prophylactic...") – this section is unclear and should be rewritten; "reduce the patient's immune system"? Note that there have been non-vaccine prophylaxis options for COVID-19 as well, e.g. Evusheld.

We understand the review's concern ("reduce the patient's immune system"?) and have attempted to provide more clarity. The intention was to convey that anti-inflammatories are administered to patients to "reduce" the impact of CRS due to a hyperactive or dysregulated immune response. We also acknowledge the reviewer's point - Evusheld was also used as a prophylactic. It has been replaced with Pengarda, a monoclonal antibody, which has maintained activity against the latest XEC variant. We have incorporated this suggestion as well. The overall intent of this paragraph is to motivate the development of a pan-CoV inhibitor.

"differential scanning fluorescence" – should read "differential scanning fluorimetry"

Thank you for bringing this to our attention. It's been corrected in the manuscript.

Final discussion paragraph, "this decoy antagonist will be immediately useful for immunocompromised patients" – understates the extensive further study and optimization needed for commercialization and clinical application.

We completely agree. As noted above, this manuscript aims to confirm an observation (e.g., symmetry) and implement a design principle (geometry and avidity). We argue we have done so and the development of a novel, bona fide clinical candidate replete with animal experiments, appropriate controls and head-to-head comparison with other modalities (ACE2-Fc) is a major undertaking and would constitute another manuscript. Regardless, we have modified the text addressing this comment.

Methods section, DSF – final concentration of SYPRO Orange is unclear.

We have added experimental details to the manuscript.

Reviewer #3 (Remarks to the Author):

Comments:

In this manuscript, Juliet Gonzales et al. sought a trimeric scaffold collagen XVIII to create an ultrapotent, trivalent ACE2 entry antagonist. Distinct disulfide bonds E31C–V'37C and N51C–V343C were added in the collagen XVIII trimerization domain and ACE2, respectively, to increase the overall thermal and serum stability. A single point mutation of R273Q was introduced to inhibit ACE2 enzyme activity. The enzymatically inactivated construct, TD3ADR273Q, closely matches the C3 symmetry of the spike. It binds to the spike with high affinity, exhibits potent viral inhibition against all tested SARS-CoV-2 variants, and shows remarkable stability in serum. TD3ADR273Q was expected to become the candidate widely effective against SARS-CoV-2 variants and other ACE2-utilizing entry viruses.

Specific comments:

1. The authors thought that the trimeric ACE2 decoy antagonist could widely inhibit SARS-CoV-2 variants and other ACE2-utilizing entry viruses. The SPR results of this paper has demonstrated that the trimeric ACE2 antagonist, TD3ADR273Q, exhibits a strong binding affinity to SARS-CoV-2 and SARS-CoV. TD3ADR273Q can inhibit the pseudovirus of SARS-CoV-2 and its variants fusion. However, there is a lack of experimental data on TD3ADR273Q's ability to neutralize SARS-CoV pseudovirus. Could you please provide additional data for this part of the experiment?

We absolutely agree and have done so. We are happy to report that the trimeric scaffold also inhibits SARS-CoV-1 using pseudoviral assays. We are grateful for the reviewer's suggestion as it confirms this approach is highly likely to be applicable as a viral inhibitor of future, unknown SARS-CoV-x strains.

2. Regarding the SPR results in Figure 1, please add the KD values to facilitate direct comparison of the binding affinities among different groups.

Thank you for the suggestion. Data was originally reported in Table 1 only, but we added these values directly to Figure 1 for clarity.

3. The researchers mention that the standard deviation ranges observed in the ACE2 monomer-bound structures are 0.8-10 Å. Is this concept widely accepted? Have you personally summarized this concept, or is it proposed in another research paper? If it is based on your own experience, please provide an explanation. If it is from a published paper, please cite the source.

Thank you for asking for clarification. We have re-written this paragraph for clarity.

We then attempted to evaluate the 'symmetry of interaction' between our ACE2 constructs with spike protein (T0A-S, T3A-S, T5A-S, and T18A-S; where S is the SARS-CoV-2 spike Hexaprotein) by likening the C3 symmetry to an equilateral triangle with all sides being equal length and a corresponding standard deviation between these lengths of zero (s.d. = 0). Thus, we calculated lengths (distances) and corresponding standard deviations between the C α atoms of equivalent residues in the RBDs of the spike protein (residue D428; as determined by Fan, et al. 2022⁷⁰) and in each of the ACE2 domains (residue N137; arbitrarily determined residue on ACE2) (Fig. 2, right, Table 3). The calculated mean distances and s.d. between spike RBD were: 41 ± 3.6 Å for T0A-S; 40 ± 1.0 Å for T3A-S; 40 ± 1.0 Å for T5A-S; and 42 ± 3.2 Å for T18A-S. Equating lower s.d. values to symmetry, we determine T3A and T5A to have higher symmetry than T0A or T18A. Next, we calculated mean distance and s.d. between ACE2: 149 ± 13.2 Å for T0A-S; 141 ± 7.6 Å for T3A-S; 144 ± 11.2 Å for T5A-S; and 142 ± 10.8 Å for T18A-S. Again, equating lower s.d. values to symmetry, we now determine T3A to have the highest symmetry of binding between constructs. These distances and s.d. were compared to those of five cryo-EM structures of spike trimer with three ACE2 monomers bound (see Methods) to compare symmetry and determine whether binding of our constructs potentially force the RBD domains into un-natural or energetically unfavorable conformations. We observed average mean inter-RBD distances of 41 ± 1.9 Å and mean inter-ACE2 distances of 144 ± 6.0 Å. As expected, the s.d. values of inter-RBD and inter-ACE2 of these models without a trimeric domain are similar or lower than any of our complexes – indicating higher order symmetry. Regardless, the mean distances between spike RBD or ACE2 in each of our constructs fall within one standard deviation of that calculated from published structures and suggests that TXA constructs do not require or induce energetically unfavorable spike RBD conformations for interactions. Combined with clear density for the collagen XVIII trimerization domain and 'optimal' positioning of individual ACE2 domains, the T3A represented the 'best' compromise and was selected for further development.